# O-methyltransferase-like enzyme catalyzed diazo installation in polyketide biosynthesis

Yuchun Zhao[1,6], Xiangyang Liu[1,6], Zhihong Xiao[1], Jie Zhou[1], Xingyu Song[2], Xiaozheng Wang[1], Lijun Hu[3], Ying Wang[3], Peng Sun[4], Wenning Wang [2], Xinyi He [1], Shuangjun Lin [1], Zixin Deng [1], Lifeng Pan[5] & Ming Jiang [1] ✉

Diazo compounds are rare natural products possessing various biological activities. Kinamycin and lomaiviticin, two diazo natural products featured by the diazobenzofluorene core, exhibit exceptional potency as chemotherapeutic agents. Despite the extensive studies on their biosynthetic gene clusters and the assembly of their polyketide scaffolds, the formation of the characteristic diazo group remains elusive. L-Glutamylhydrazine was recently shown to be the hydrazine donor in kinamycin biosynthesis, however, the mechanism for the installation of the hydrazine group onto the kinamycin scaffold is still unclear. Here we describe an O-methyltransferase-like protein, AlpH, which is responsible for the hydrazine incorporation in kinamycin biosynthesis. AlpH catalyses a unique SAM-independent coupling of L-glutamylhydrazine and polyketide intermediate via a rare Mannich reaction in polyketide biosynthesis. Our discovery expands the catalytic diversity of O-methyltransferase-like enzymes and lays a strong foundation for the discovery and development of novel diazo natural products through genome mining and synthetic biology.

N−N bond-containing natural products (NPs) are a group of important molecules with enormous structural diversities. More than 300 such NPs with different N−N bonds, including diazo, hydrazide, azoxy, hydrazine, hydrazone, and pyridazine, have been discovered and characterized in the past few decades[1]. These NPs attract numerous research attentions due to their structural diversities and broad spectrum of biological activities, including antifungal, antiviral, antitumor, and other activities[2]. In contrast, the biosynthetic machineries of N−N bond formation remain to be elucidated until recently. Over the past ten years, our understanding of enzymatic N−N bond formation has significantly improved due to the uncovering of an increasing number of biosynthetic gene clusters (BGCs) for N−N bond containing NPs[3–11]. The nitrogen donors and the mechanisms for the installation of

the N−N bonds in those NPs have been elucidated through detailed genetic and biochemical studies. To overcome the challenges in generating N−N bond chemically, nature has evolved several different biosynthetic strategies, including the activation of amine through an electrophilic intermediate, utilizing nitrite as a nitrogen donor, and radical-mediated nonenzymatic mechanism[1,12].

The diazo group is an important functional group in organic synthesis but rarely presents in NPs (Fig. 1a). Previous biosynthetic studies in diazo group formation suggested a stepwise nitrogen incorporation involving a diazotization process[13]. Nitrate has long been proposed to be involved in quite a few N−N bond-containing NP biosynthetic processes. Recent studies in the diazoquinone NP, cremeomycin, revealed an unprecedented nitrous acid biosynthetic pathway

[1]State Key Laboratory of Microbial Metabolism, Joint International Research Laboratory of Metabolic & Developmental Sciences, and School of Life Sciences and Biotechnology, Shanghai Jiao Tong University, 200030 Shanghai, P. R. China. [2]Ministry of Education Key Laboratory of Computational Physical Sciences, Department of Chemistry, Institutes of Biomedical Sciences, Fudan University, 200438 Shanghai, China. [3]Guangdong Province Key Laboratory of Pharmacodynamic Constituents of TCM and New Drugs Research, Center for Bioactive Natural Molecules and Innovative Drugs Research, Jinan University, 510632 Guangzhou, P. R. China. [4]School of Pharmacy, Second Military Medical University, 325 Guo-He Road, 200433 Shanghai, P. R. China. [5]State Key Laboratory of Bioorganic and Natural Products Chemistry, Shanghai Institute of Organic Chemistry, University of Chinese Academy of Sciences, Chinese Academy of Sciences, 200032 Shanghai, China. [6]These authors contributed equally: Yuchun Zhao, Xiangyang Liu. ✉e-mail: jiangming9722@sjtu.edu.cn

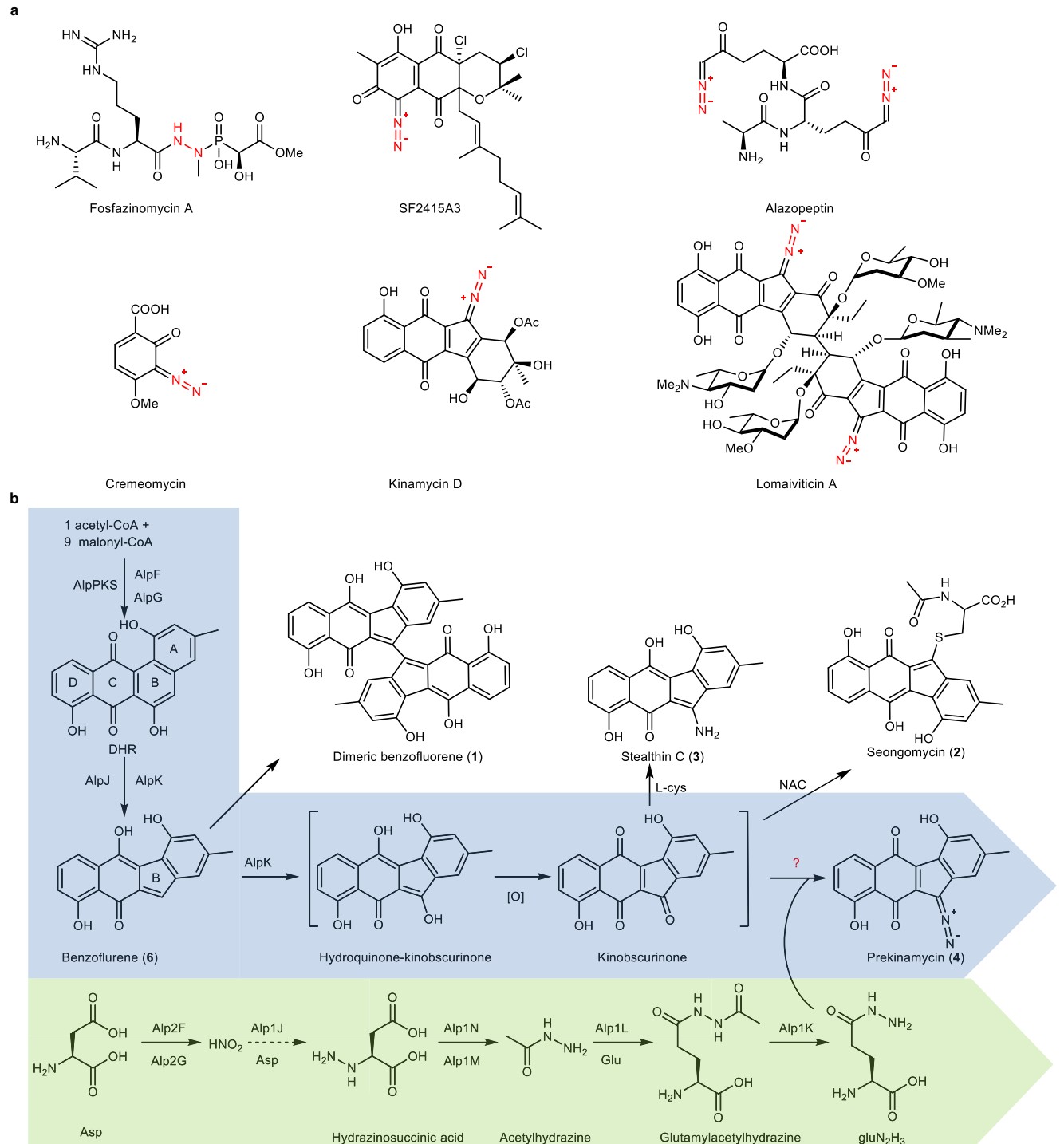

**Fig. 1 | Selected natural products containing nitrogen–nitrogen bonds and key steps of the prekinamycin biosynthesis. a** Selected natural products containing nitrogen–nitrogen bonds. The nitrogen–nitrogen bonds are highlighted in red. **b** Proposed pathway for the biosynthesis of prekinamycin. Proposed pathways for the biosynthesis of polyketide and gluN$_2$H$_3$ are highlighted in blue and green.

responsible for the formation of the diazo group[14,15]. Nitrous acid is generated from aspartic acid catalyzed by a flavin-dependent mono-oxygenase (CreE) and a nitrosuccinate lyase (CreD), and subsequently reacts with the primary aromatic amine of 3-amino-2-hydroxy-4-methoxybenzoic acid to form the diazo group spontaneously in an acidic condition or enzymatically by an ATP-dependent diazo-forming protein, CreM.

The kinamycins and lomaiviticins form a small group of NPs featuring diazobenzoluorene core structures[16]. Their biosynthesis is of

intense interest due to their impressive biological activities[17]. The BGCs of kinamycin[18–20] and lomaiviticin[21] have been characterized, and the heterologous production of kinamycin propels the genetic and bio-chemical studies of the biosynthesis of kinamycin family NPs. The biosynthetic machinery for constructing the benzofluorene skeleton has been proposed and biochemically verified[22] (Fig. 1b). During the assembly of the benzofluorene core, two FAD-dependent enzymes, AlpJ and AlpK, were identified to be involved in the contraction of the B ring in the kinamycin biosynthesis[22, 23]. AlpJ belongs to a unique family of

FAD-dependent oxygenase and has been shown to catalyze a Bayer-Villiger-type oxidative C-C bond cleavage of dehydrorabelomycin (DHR), forming of an unstable aldehyde/acid intermediate, which undergoes decarboxylative ring closure and dehydration to give the benzofluorene ring. AlpK could possibly supply necessary FADH$_2$ for AlpJ catalyzed reaction. And it was also proposed that AlpK might catalyze the C5-hydroxylation reaction of the benzofluorene intermediate. Meanwhile, the enzymes involved in the late stage tailoring steps, such as the acetylation[18] and epoxide hydrolyzation[20] on A ring, have been identified and characterized. However, there is still a knowledge gap in the biosynthesis of the diazo group in kinamycin and lomaiviticin. Previous feeding experiments suggested that kinobscurinone[13] and stealthin C[24] isolated from *Streptomyces murayamaensis* mutants are the biosynthetic intermediates of kinamycin, indicating a step-by-step nitrogen incorporation in diazo formation analogous to the biosynthetic scenario of cremeomycin[15] and SF2415A3[25]. However, this hypothesis is challenged by recent studies. The biosynthetic studies of the N−N bond formation in two structurally unrelated NPs, fosfazinomycin and kinamycin, provided more information about the details of the N−N bond formation in kinamycin family NPs. The N−N bond in kinamycin was confirmed to be constructed separately, followed by incorporation into the polyketide scaffold through a convergent biosynthetic process reminiscent of the strategy used in organic synthesis[26]. Particularly, L-glutamylhydrazine (gluN$_2$H$_3$) possessing the N−N bond synthon was identified as the essential biosynthetic intermediate in kinamycin biosynthesis through in-depth biochemical assays and feeding experiments.

Except for the N−N bond formation step, gluN$_2$H$_3$ biosynthesis has been well characterized[26] (Fig. 1b). Briefly, aspartic acid is oxidized by FzmM/Alp2F to generate nitrosuccinate, which is cleaved by FzmL/Alp2G to release nitrous acid. Subsequently, the nitrous acid is proposed to be incorporated into aspartic acid through the N−N bond to generate hydrazinosuccinic acid by FzmP/Alp1J which still lacks biochemical and genetic evidence. The hydrazinosuccinic acid is acetylated by FzmQ/Alp1N to produce N-acetylhydrazinosuccinic acid, which is cleaved by FzmR/Alp1M to release acetylhydrazine. Eventually, the acetylhydrazine is transferred onto glutamic acid by FzmN/Alp1L to produce glutamylacetylhydrazine, which is further deacetylated to generate gluN$_2$H$_3$ by FzmO/Alp1K. However, the enzyme(s) and mechanism involved in the installation of gluN$_2$H$_3$ or the hydrazine onto the polyketide scaffold remain unknown. The putative enzyme responsible for the coupling reaction is ambiguous in kinamycin BGC. A conserved amidotransferase-like enzyme, Alp1W, was proposed to transfer the hydrazine synthon to the polyketide scaffold without any genetic or biochemical evidence[26].

In this work, we reconstitute the production of prekinamycin (**4**), the known inline diazo-containing intermediate in kinamycin biosynthesis, in vitro successfully for the first time and characterize an important benzofluorene intermediate, compound **5**, a condensation product of gluN$_2$H$_3$ and the polyketide intermediate (**7**). Particularly, the *O*-methyltransferase (OMT)-like enzyme, AlpH, from the kinamycin BGC is determined to mediate the C-N bond formation through loading gluN$_2$H$_3$ to the polyketide scaffold. In vitro studies show that compound **5** is produced from DHR through a cooperative catalysis by AlpJ, AlpK, and AlpH. This study provides significant insights into the installation of the diazo group in kinamycin family NPs, a biomarker for in silico genome mining of putative diazo NPs and a biobrick to generate novel diazo NPs through synthetic biology.

## Results

### In vivo genetic verification of *alpH*'s role in kinamycin biosynthesis

Although the BGC for kinamycin has been identified by us and other groups separately, the putative enzyme for the N−N bond installation is still obscure. A previous study predicts that Alp1W (a putative

glutamine amidotransferase) as the putative enzyme to transfer hydrazine onto kinobscurinone, which is produced from DHR by AlpJ and AlpK, with no experimental evidence[26]. Alp1W is conserved among kinamycin, lomaiviticin, and nenestatin biosynthetic pathways, which all include diazo incorporation. However, there is no *alp1W* homologous gene in the BGC of fluostatin, whose biosynthesis also includes a similar diazo incorporation (Supplementary Fig. 1). To test the function of Alp1W in vivo, we generated an *alp1W* in-frame null mutant strain, *S. albus* J1074/W-3C2Δ*alp1W*. The Δ*alp1W* strain lost most of its ability to produce kinamycin and no other related compounds were accumulated according to the high-performance liquid chromatography (HPLC) profile (Supplementary Fig. 2), indicating the essential role of Alp1W in kinamycin biosynthesis. The production of kinamycin was restored to a comparable level to the wild-type strain by ex-situ complementation of *alp1W* driven by its own promoter. However, we can't tell whether Alp1W is involved in the gluN$_2$H$_3$ loading onto the polyketide scaffold. To further test the aforementioned hypothesis in vitro, Alp1W, AlpJ, and AlpK were expressed, and purified from *Streptomyces* or *Escherichia coli* and incubated with DHR, NADH, and gluN$_2$H$_3$ in a one-pot reaction (Supplementary Figs. 3, 4). As seen in a previous AlpJK one-pot reaction, the dimeric benzofluorene product (**1**) was generated and confirmed by liquid chromatography coupled to high-resolution mass spectrometry (LC-HRMS) (Supplementary Fig. 5). Unfortunately, we could not detect any other new products after elaborate attempts. Apparently, our data challenges the hypothesis of Alp1W as the N−N bond installation enzyme in the kinamycin biosynthesis.

However, there is an OMT family protein coding gene, *alpH*, present in all kinamycin and lomaiviticin family NP BGCs with unassigned function in their biosynthesis (Supplementary Fig. 1). AlpH appears highly conserved in kinamycin, lomaiviticin, fluostatin, and nenestatin biosynthetic pathways (Supplementary Figs.1 and 6, Supplementary Table 3). It possesses an N-terminal glycine rich motif, DFCGGQG, suggesting a putative SAM binding motif (DxGxGxG)[27] (Supplementary Fig. 6). AlpH also exhibits moderate homology to the OMT, MmcR[28] (89% coverage/55% identity), from the mitomycin BGC. Interestingly, *O*-methylation seems to be unnecessary for the kinamycin biosynthesis (Supplementary Fig. 7). However, we could not exclude the possibility of a cryptic methylation in the kinamycin biosynthesis, which has been reported in other NPs' biosynthesis such as the C-terminal methylation in thiostrepton biosynthesis[29]. Additionally, kinafluorenone[30], an *O*-methylated shunt product from the kinamycin biosynthesis was also isolated from the *S. murayamaensis* mutant strain MC1. To verify the function of AlpH in kinamycin biosynthesis, we generated an *alpH* in-frame deletion mutant strain, *S. albus* J1074/W-3C2Δ*alpH*. Kinamycin production was completely abolished in this Δ*alpH* strain and was restored to a comparable level to the wild-type strain by ex-situ complementation of *alpH* driven by the *kasO*p* promoter (Fig. 2a). Despite the disappearance of kinamycin D peak from the HPLC chromatogram of Δ*alpH* strain, two new peaks corresponding to seongomycin (**2**) and stealthin C (**3**) were observed, which were further confirmed by comparison with the authentic standards using HPLC and HRMS (Supplementary Figs. 8, 9). **2** and **3** are both shunt products from the kinamycin biosynthesis derived from DHR and converted by AlpJ (AlpK may also be involved)[23]. Accumulation of **2** and **3** in Δ*alpH* strain indicates that AlpH is not required for the construction of the polyketide scaffold and may function after the AlpJK-mediated ring contraction in kinamycin biosynthesis. As observed in Δ*alpH* strain, the accumulation of **2** was also seen in our previous Δ*alp2F-2G* strain. Alp2F and Alp2G are homologs of CreE and CreD, catalyzing the generation of nitrous acid from aspartic acid[18]. In kinamycin biosynthesis, Alp2F and Alp2G are required for the biosynthesis of gluN$_2$H$_3$, which is the N−N bond carrier. However, the biosynthesis of gluN$_2$H$_3$ has not been fully characterized, and the key enzymes involved in the N−N bond formation have not been

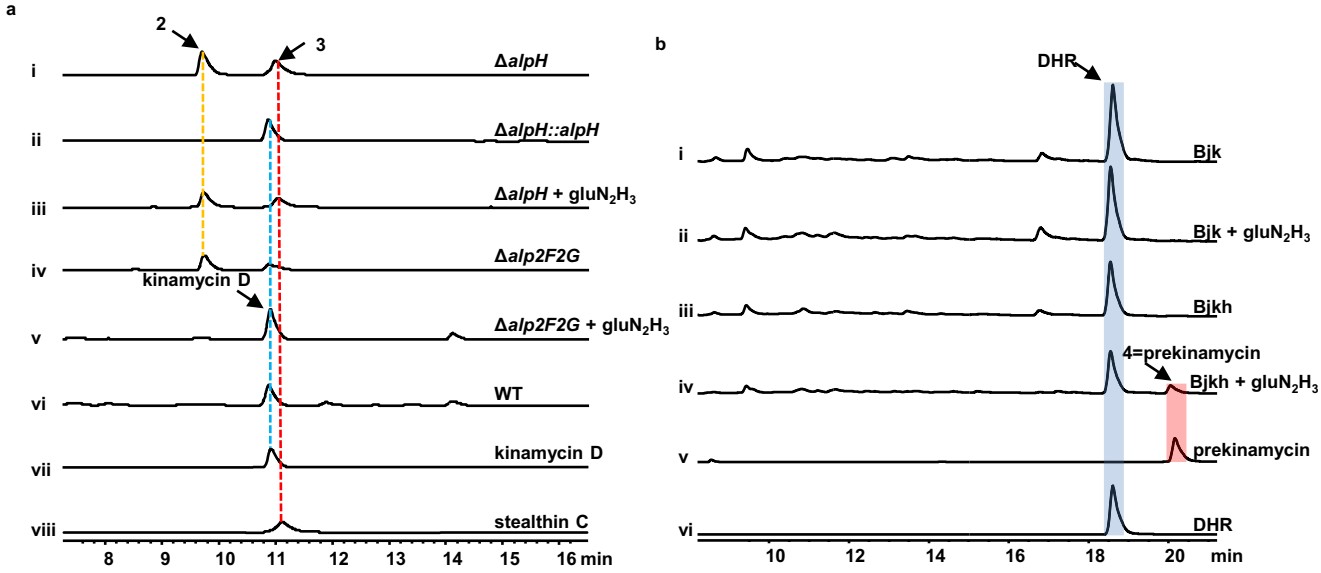

**Fig. 2 | In vivo functional analysis of AlpH. a** HPLC profiles of the crude extracts from different strains: (i) ΔalpH strain; (ii) ΔalpH::alpH strain; (iii) ΔalpH strain fed with gluN₂H₃; (iv) Δalp2F2G strain; (v) Δalp2F2G strain fed with gluN₂H₃; (vi) wild-type strain; (vii) kinamycin D standard; (viii) stealthin C standard. **b** HPLC profiles of the crude extracts from the *E. coli* BL21(DE3) derivatives fed with gluN₂H₃: (i) DHR producing strain plus *alpJ-K* (Bjk); (ii) Bjk fed with gluN₂H₃; (iii) DHR producing strain plus *alpJ-K-H* (Bjkh); (iv) Bjkh fed with gluN₂H₃; (v) prekinamycin standard; (vi) DHR standard.

determined either. To test if AlpH is involved in gluN₂H₃ biosynthesis, synthetic gluN₂H₃ was fed to *ΔalpH* strain. In contrast to *Δalp2F-2G* strains, gluN₂H₃-fed *ΔalpH* strain couldn't restore the production of kinamycin D, suggesting that AlpH is independent from gluN₂H₃ biosynthesis (Fig. 2a). It is noteworthy that both mutant strains exhibited similar HPLC metabolic profile. Collectively, AlpH may play a role in the biosynthesis steps between the benzofluorene intermediate (**6**) and prekinamycin and be involved in the installation of the N−N bond to the polyketide scaffold. Since AlpH is predicted to be an OMT, it may catalyze a cryptic methylation, which is essential for subsequent installation of the N−N bond on the polyketide intermediate generated by AlpJ and AlpK. The methyl group may be removed during the maturation of prekinamycin. Alternatively, AlpH could probably function as a gluN₂H₃ transferase instead of an OMT to install the N−N bond synthon onto kinamycin polyketide scaffold. Overall, the in vivo data suggests that AlpH plays an important role in kinamycin biosynthesis.

**Reconstitution of diazo formation in kinamycin biosynthesis**
We've developed an *E. coli*-based expression system for producing DHR in previous studies, providing a robust heterologous platform to investigate the function of AlpH in kinamycin biosynthesis[31]. Given that AlpH may participate in the biosynthesis after AlpJ and AlpK, it was co-expressed in the DHR-producing strain with AlpJ and AlpK, resulting in Bjkh strain. Unfortunately, we did not see any differences between the control strain and the Bjkh strain from the HPLC metabolic profile (Fig. 2b). We speculated that the AlpH-catalyzed product may be unstable. Subsequently, we fed gluN₂H₃ into Bjkh culture and observed the appearance of a new product (**4**) from the HPLC chromatogram. **4** exhibited an identical UV absorbance spectrum to prekinamycin, indicating the formation of a diazo product (Supplementary Fig. 10a). Large scale fermentation and purification yielded analytically pure **4** with a mass/charge ratio of 317.0568 ([M−H]⁻) determined by HRMS, identical to that of prekinamycin (Supplementary Fig. 10b). **4** was proven to be prekinamycin through further comparison to the authentic standard using HPLC (Fig. 2b). The production of **4** in Bjkh strain fed with gluN₂H₃ suggests that AlpH is involved in the diazo formation in prekinamycin and can convert DHR to prekinamycin in cooperation with AlpJ and AlpK in the presence of gluN₂H₃.

We next sought to reconstitute the function of AlpH in vitro. AlpH was cloned, expressed, and purified into homogeneity from *E. coli* as N-terminal His₆-tagged protein (Supplementary Fig. 3). The above in vivo data suggest that the substrate for AlpH is the product of DHR catalyzed by AlpJ and AlpK, which has been well-known to be unstable. Therefore, we designed a one-pot coupled assay using DHR as the substrate in the presence of AlpJ, AlpK, gluN₂H₃, FAD, NADH, and *S*-adenosylmethionine (SAM). HPLC monitoring of this one-pot reaction revealed the production of prekinamycin (Fig. 3a). Meanwhile, prekinamycin was absent in the control reactions using boiled AlpH or without gluN₂H₃, indicating that the diazo installation is AlpH and gluN₂H₃-dependent. Taken together, AlpJ, AlpK, and AlpH were sufficient to convert DHR to prekinamycin using gluN₂H₃ as the N−N bond donor.

To further ascertain the details of the transformation of DHR into prekinamycin, we analyzed the reaction carefully and monitored the one-pot reaction in a time-course manner, which led to the discovery of a new product (compound **X**) beside prekinamycin (Fig. 3b). Since **X** is unstable during solvent extraction, the one-pot reaction was directly injected into HPLC for analysis without further clean-up. **X** was massively produced in 35 min and then gradually decreased with the appearance of prekinamycin after 120 min in the reaction, suggesting **X** as the true turnover product and prekinamycin as the decomposition product in the one-pot reaction. **X** exhibited a distinct UV absorbance spectrum from prekinamycin and DHR, and its molecular weight was determined to be 450.1296 ([M+H]⁺) through LC-HRMS, corresponding to a molecular formula of C₂₃H₁₉N₃O₇ (Supplementary Fig. 11a, Fig. 3c). During the purification of **X** through HPLC, we noticed its spontaneous conversion to prekinamycin and a compound with the same molecular weight of dehydrated glutamic acid in DMSO (Supplementary Fig. 11b–d). MS/MS fragmentation of **X** yielded two major daughter ions, *m/z* 321.0881 and *m/z* 130.0503, which are consistent with the products of "dihydroprekinamycin" and glutamyl ions (Fig. 3d). Therefore, **X** was determined to be **5**, which is the adduct of gluN₂H₃ onto the polyketide scaffold. When any one of these three enzymes and NADH was removed from the assay, **5** was no longer produced, indicating all three enzymes and NADH are essential for the conversion of DHR to **5** (Supplementary Fig. 12). In addition, we noticed the production of benzofluorene (**6**) and dimeric

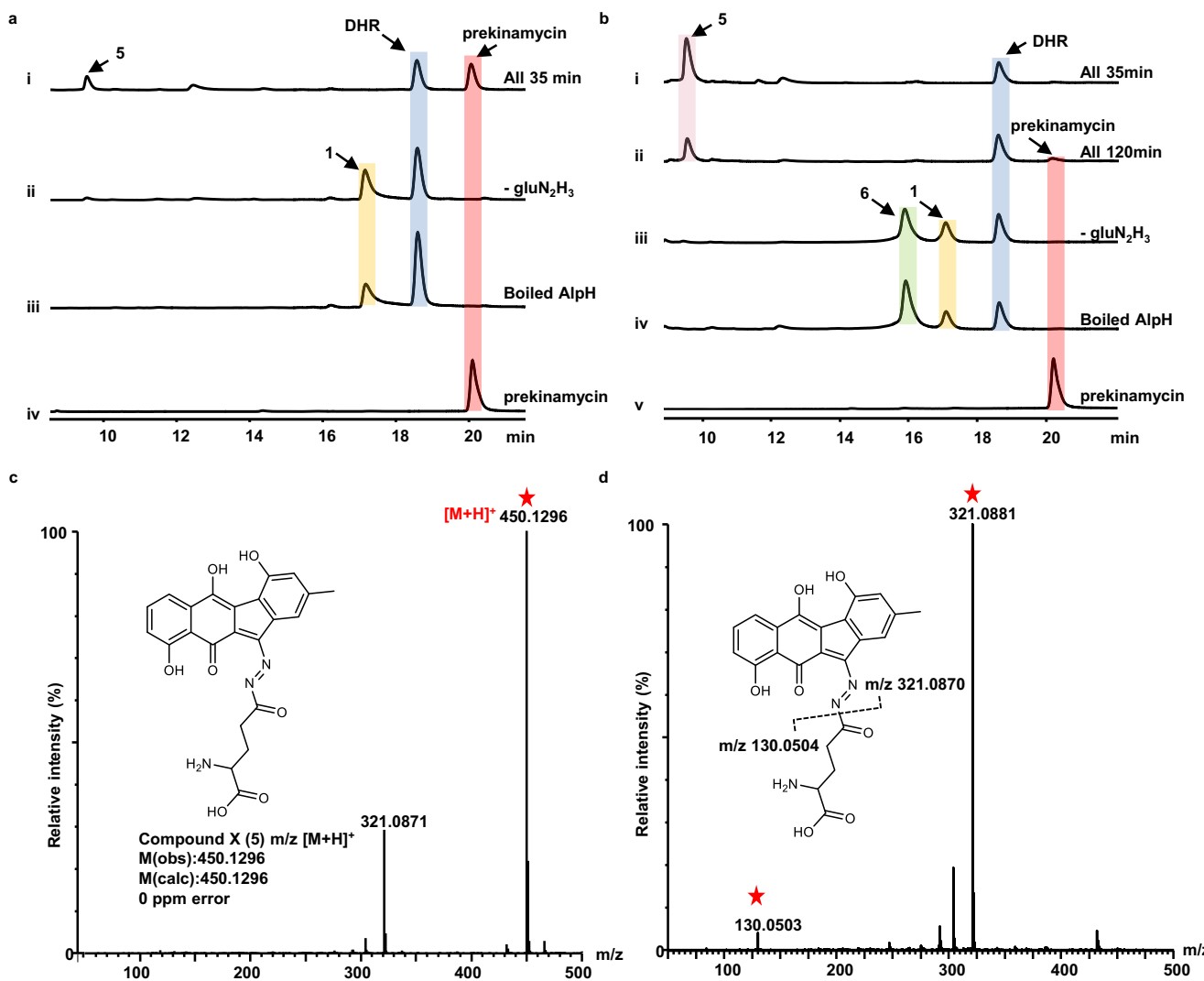

**Fig. 3 | In vitro functional characterization of AlpH. a** HPLC analysis of the organic extracts from the enzymatic reactions: (i) DHR + AlpJ + AlpK + AlpH + NADH + gluN$_2$H$_3$; (ii) DHR + AlpJ + AlpK + AlpH + NADH; (iii) DHR + AlpJ + AlpK + boiled AlpH + NADH + gluN$_2$H$_3$; (iv) prekinamycin standard. (**b**) Direct HPLC analysis of the enzymatic reactions: (i) DHR + AlpJ + AlpK + AlpH + NADH + gluN$_2$H$_3$ (35 min of reaction time); (ii) DHR + AlpJ + AlpK + AlpH + NADH + gluN$_2$H$_3$ (120 min of reaction time); (iii) DHR + AlpJ + AlpK + AlpH + NADH; (iv) DHR + AlpJ + AlpK + boiled AlpH + NADH + gluN$_2$H$_3$; (v) prekinamycin standard. **c** LC-HRMS data (positive mode) of **5**. **d** MS/MS (positive mode) spectrum and proposed fragmentation of **5**.

benzofluorene (**1**) in the control reactions, which were also observed by Yang and co-authors[22] (Supplementary Fig. 13). Our results unambiguously rule out the possibility of hydrazine, the hydrolyzed product of gluN$_2$H$_3$, as the N−N bond donor in kinamycin biosynthesis. Instead of hydrazine, gluN$_2$H$_3$ was directly loaded onto the kinamycin biosynthetic intermediate, which may possess a carbonyl group to accept the gluN$_2$H$_3$ adduct, generating **5**. **5** can be further hydrolyzed and oxidized into prekinamycin spontaneously at room temperature.

Intriguingly, we didn't identify any methylated intermediates in either the *E. coli* in vivo reconstitution system or the in vitro one-pot reaction system, which is contradictory to the predicted function of AlpH as a methyltransferase. Therefore, we removed SAM from the in vitro one-pot reaction for further analysis. Surprisingly, the production of **5** was significantly improved without SAM (Supplementary Fig. 14). In other words, the production of **5** was significantly inhibited by exogenously added SAM. Similarly, the production of **5** was significantly decreased by two SAM analogues, SAH and sinefungin (Supplementary Fig. 14). To rule out the carryover of SAM from protein purification, AlpH was denatured, and no SAM was detected in the supernatant by LC-MS (Supplementary Fig. 15). To further investigate whether SAM is required for AlpH activity, G203A mutation was introduced into its putative SAM binding motif. AlpH G203A can still mediate the production of **5** efficiently in the in vitro one-pot reaction (Supplementary Fig. 16). Therefore, the results hereby indicate that, although AlpH shows significant homology to OMT, SAM is not required for its functionalization. Phylogenetic analysis of AlpH with methyltransferases from other NP BGCs showed that AlpH and its homologs form a separate clade, indicating their distinct functions (Supplementary Fig. 17, Supplementary Table 4).

## AlpH catalyzes N−N incorporation in kinamycin biosynthesis

Since AlpH is a SAM-independent enzyme, we propose that AlpH may catalyze the transfer of gluN$_2$H$_3$ onto the polyketide scaffold. However, we can't rule out the possibility of AlpH catalyzing an unknown modification on the polyketide product generated by AlpJ and AlpK from DHR, which can facilitate the transfer of gluN$_2$H$_3$. It has been reported that seongomycin (**2**) was produced by a spontaneous nucleophilic attack of N-acetylcysteine (NAC) to an electrophilic intermediate, putatively hydroquinone-kinobscurinone, converted from DHR by AlpJ and AlpK[23]. We hypothesize that the production of **2** will be impaired if AlpH modifies the product/intermediates generated by AlpJ and AlpK. Therefore, AlpH was added to the one-pot reaction mixture for

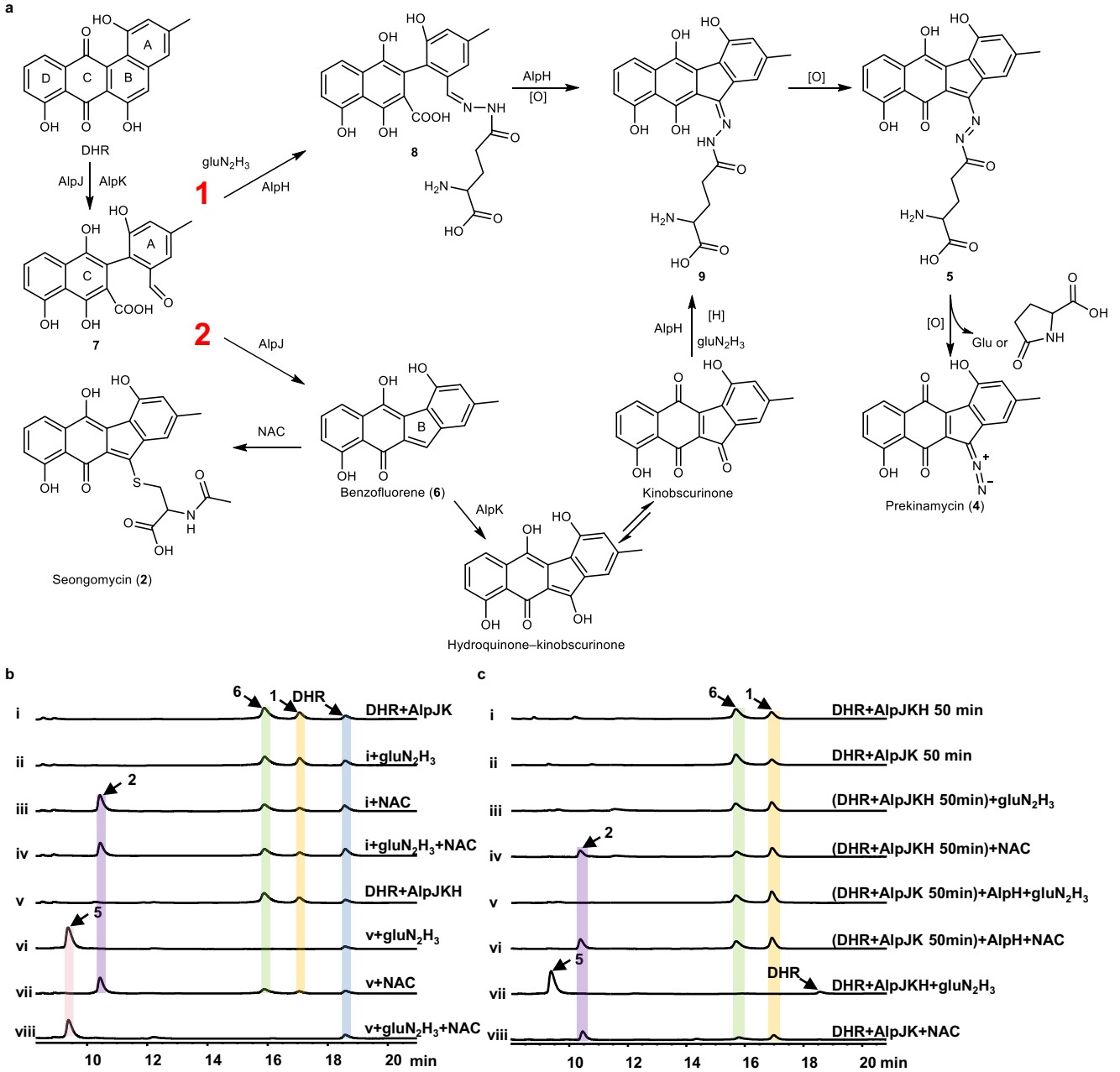

**Fig. 4 | Mechanisms of gluN₂H₃ and NAC installations onto the polyketide skeleton. a** Mechanistic alternatives for the coupling of gluN₂H₃ and polyketide intermediate. **b** HPLC analysis of the competition reactions: (i) DHR + AlpJ + AlpK + NADH; (ii) DHR + AlpJ + AlpK + NADH + gluN₂H₃; (iii) DHR + AlpJ + AlpK + NADH + NAC; (iv) DHR + AlpJ + AlpK + NADH + gluN₂H₃ + NAC; (v) DHR + AlpJ + AlpK + AlpH + NADH; (vi) DHR + AlpJ + AlpK + AlpH + NADH + gluN₂H₃; (vii) DHR + AlpJ + AlpK + AlpH + NADH + NAC; (viii) DHR + AlpJ + AlpK + AlpH + NADH + gluN₂H₃ + NAC. **c** HPLC analysis of the two-step reactions: (i) DHR + AlpJ + AlpK + AlpH +

NADH, DHR was consumed with the formation of **1** and **6**; (ii) DHR + AlpJ + AlpK + NADH, DHR was consumed with the formation of **1** and **6**; (iii) gluN₂H₃ was added to reaction (i) when DHR was totally consumed; (iv) NAC was added to reaction (i) when DHR was totally consumed; (v) AlpH and gluN₂H₃ was added to reaction (ii) when DHR was totally consumed; (vi) AlpH and NAC was added to reaction (ii) when DHR was totally consumed; (vii) DHR + AlpJ + AlpK + AlpH + NADH + gluN₂H₃; (viii) DHR + AlpJ + AlpK + NADH + NAC.

producing **2** including AlpJ, AlpK, DHR, and NAC. There was no difference in the production of **2** between the control reaction and the AlpH-added reaction from the HPLC chromatograms (Supplementary Fig. 18), indicating that AlpH can't modify the product/intermediates generated by AlpJ and AlpK. Collectively, AlpH is most likely catalyzing the transfer of gluN₂H₃ directly onto the polyketide scaffold, producing **5**, in kinamycin biosynthesis.

There are two different mechanisms to generate **5** from DHR, 1) a Mannich-like reaction-mediated gluN₂H₃ condensation with an aldehyde intermediate (**7**) similar to the isoleucine coupling in jadomycin

biosynthesis and ring closure through an enolate anion or hydroquinone attack on the iminium[32,33]; 2) coupling of gluN₂H₃ to the polyketide scaffold after ring contraction, featuring a putative benzofluorene intermediate, which is in alignment with the formation of **2** (Fig. 4a). In mechanism 2, **2** and **5** compete for the same polyketide substrate. To further investigate the underlying mechanism of AlpH, competition assays were performed using the one-pot reaction mixture including AlpJ, AlpK AlpH, and DHR in the presence of gluN₂H₃ and NAC. When equal amounts of gluN₂H₃ and NAC (molar ratio 1:1) were added, only **5** but not **2** was detected in the one-pot reaction (Fig. 4b). The

supplementation of 10 times more NAC over $gluN_2H_3$ still did not result in the production of **2** (Supplementary Fig. 19), suggesting that AlpH may utilize a distinct polyketide substrate to generate **5**. We also noticed that **1** and **6** are only produced in the one-pot reaction for producing **2** but not **5**, indicating that the coupling of $gluN_2H_3$ may proceed before the closure of the five-membered ring B (mechanism 1). To verify this hypothesis, we tested the activity of AlpH towards the putative five-membered ring intermediates, including **6** and its hydroxylated product hydroquinone-kinobscurinone. Due to their well-known instability, **6** and hydroquinone-kinobscurinone were prepared in situ by incubating AlpJ and AlpK with DHR and NADH as previously reported. AlpH and $gluN_2H_3$ were added to the reaction after complete conversion of DHR into **1** and **6** ($t = 50$ min). However, no production of **5** was observed. In contrast, **2** was produced to a level similar to the control reactions with NAC added at $t = 0$ min and $t = 50$ min (Fig. 4c). Furthermore, when isoleucine was added to the reaction mixture without AlpH instead of $gluN_2H_3$ jadomycin A could be detected by LC-MS indicating that **7** is the potential product of AlpJ and AlpK catalyzed reaction (Supplementary Fig. 20). Therefore, although **2** and **5** share the same precursor, DHR, and possess identical benzofluorene scaffold, their direct precursors are different. Taken together, we propose that AlpH exploits a decarboxylative Mannich reaction to incorporate the N−N bond and construct the five-membered ring B in kinamycin biosynthesis (Fig. 4a). After the formation of **5**, a hydrolytic release of glutamic acid followed by oxidation completes the installation of the diazo group in kinamycin, resulting in prekinamycin.

## Crystal structure of AlpH

We determined the crystal structure of apo-AlpH to 1.87 Å resolution using the molecular replacement method (Supplementary table 5), which exhibits a symmetric dimer conformation (Fig. 5a). Each AlpH monomer possesses 16 α-helixes (α1-α16) and 9 β-strands (β1−β9), forming two distinct sub-domains: an N-terminal dimerization domain and a C-terminal canonic α/β Rossmann fold catalytic domain. The swapped dimer conformation is mediated by the N-terminal dimerization domain of each AlpH monomer, which interlocks with each other and forms an extensive dimer interface, covering a total surface area of 3947 Å$^2$ (Fig. 5b). Except for the protruding N-terminal α−helix region, the overall architecture of AlpH monomer/dimer is highly similar to that of the SAM-dependent OMT LaPhzM[34] (PDB ID: 6C5B; RMSD, 2.155 Å) and MmcR[28] (PDB ID: 3GWZ; RMSD, 2.188 Å) (Supplementary Fig. 21a−d), and the SAM-independent OMT-like pericyclase PdxI[35] (PDB ID: 7BQK; RMSD, 2.512 Å) (Supplementary Fig. 21e, f) that catalyzes a unique Alder-ene reaction in fungal NP biosynthesis, as revealed by a structural similarity search using the program Dali[36]. Structural comparison analyses of AlpH with LaPhzM and MmcR revealed an extra protruding N-terminal α−helix in AlpH that can further stabilize the interlocked dimer structure (Supplementary Fig. 21a, c). However, PdxI possesses one more N-terminal α−helix comparing to AlpH for the formation of the swapped dimer (Supplementary Fig. 21e). Notably, although the overall structure of the catalytic domain of AlpH shows high similarity to the SAM-dependent OMT LaPhzM and MmcR, there is no incorporation of SAM in the determined AlpH structure, in agreement with the previous finding that purified AlpH is free of SAM or SAH.

Further structural comparison identified a unique solvent-exposed and negative charge enriched pocket between the dimerization domain and the catalytic domain in AlpH, which corresponds to the SAM/SAH binding pocket in MmcR[28] (Fig. 5c, Supplementary Fig. 22a−c). Sequence alignment and structural comparison of the

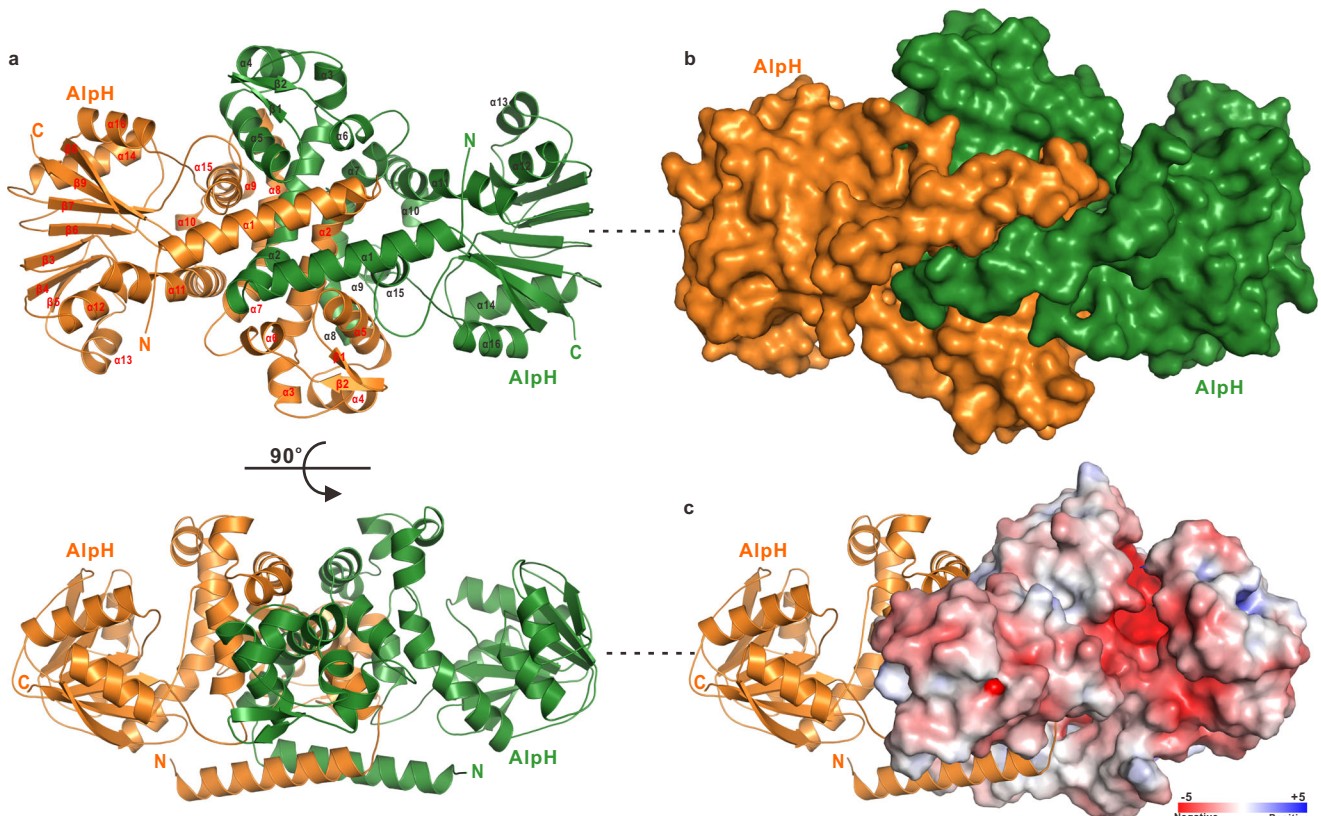

**Fig. 5 | The crystal structure of AlpH dimer. a** Ribbon diagram showing the overall structure of AlpH dimer. In this drawing, the two monomeric AlpH are colored in orange and forest green, respectively. **b** Surface representation showing the overall structure of AlpH dimer with the same orientation and color scheme as in panel A.

**c** The combined ribbon diagram and surface charge potential representation (contoured at ±5 kT/eV; blue/red) indicates the putative negatively charged substrate-binding pocket of AlpH.

SAM/SAH binding pocket revealed that the critical glycine-rich GxGxG motif and F241 residue for SAH binding in MmcR/SAH complex are replaced by CxGxG and L252 residue in AlpH (Supplementary Figs. 6, 22d, e). Apparently, the bulky side chain of Cys introduces steric hindrance for SAM/SAH binding and the replacement of F241 by L252 also eliminates the putative π-π stacking interaction between SAM/SAH and the aromatic side chain of F241, leading to the reduced binding ability of AlpH for SAM/SAH.

Given the uniqueness of the solvent-exposed pocket in AlpH, we propose that this pocket might be the substrate-binding pocket to generate **9**. We performed a molecular docking analysis of AlpH with **9** due to its relatively rigid structure comparing to the two flexible substrates. A search for the best complementary shape and the lowest binding energy led to a reasonable conformer of **9**, which accommodates the pocket very well and forms reasonable hydrophobic and polar interactions with the surrounding residues in AlpH (Fig. 6a–c)

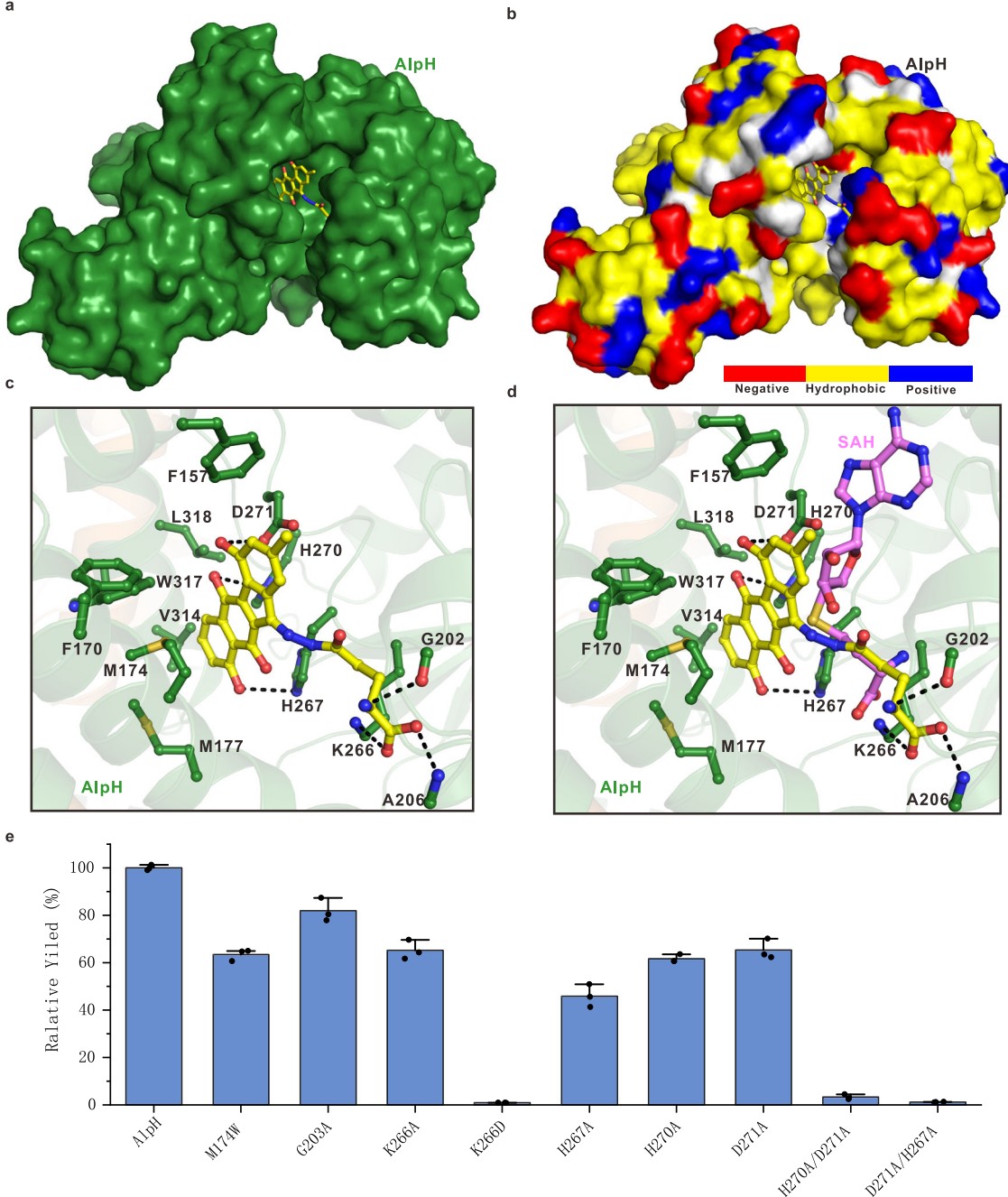

**Fig. 6 | The docking model of AlpH in complex with the product 9.**
**a** Combination of a surface representation and the stick-ball model showing the overall docking model of a monomeric AlpH in complex with the product **9**. The fitted product **9** is shown in stick model (orange) and is embedded in the pocket between the N-terminal dimerization domain and the C-terminal catalytic domain. **b** Surface representation showing the overall docking model of a monomeric AlpH in complex with the product **9** with the same orientation as in panel A. Specifically, the amino acid residues were colored as yellow (hydrophobic amino acid), blue (positively charged amino acid), red (negatively charged amino acid), and gray (uncharged polar amino acid). **c** The combined ribbon diagram and the stick-ball model showing the detailed interactions between AlpH and the fitted product **9**. The related hydrogen bonds and salt bridges involved in the binding are shown as dotted lines. **d** A structural modeling analysis of the product **9** and SAH in binding to AlpH revealing that AlpH is unable to simultaneously interact with the product **9** and SAH. **e** Relative activities of AlpH mutants. The columns represent the average values of the products (the highest mean value was set 100%); bars indicate ± SD (standard deviation) of $n = 3$ independent replicates. Source data are provided as a Source Data file.

and was confirmed by MD simulations (Supplementary Fig. 23). In particular, the benzofluorene ring of **9** fits into a hydrophobic pocket formed by the side chains of F157, F170, M174, M177, V314, W317, and L318 (Fig. 6c). Hydrogen bonds are also observed between the three hydroxy groups of ring A, B, and C of **9** and the side chains of D271, H270 and H267. In addition, the glutamyl side chain of **9** forms a salt bridge with the side chain of K266 and a hydrogen bond with the carbonyl group of G202. Sequence alignment analysis indicate all these amino acid residues potentially involved in the substrate binding in the active site of AlpH are conserved through AlpH and its homologous proteins (Supplementary Fig. 6). A further molecular docking attempt using SAM/SAH revealed that AlpH cannot adopt **9** and SAM/SAH simultaneously, owing to putative steric hindrances (Fig. 6d). To further probe their functions in catalysis, we used site-directed mutagenesis to introduce M174W, K266A, K266D, H267A, H270A, and D271A mutations into AlpH separately which were overexpressed and purified (Supplementary Fig. 24). The M174W, K266A, K266D, H267A, H270A, and D271A mutations impair the catalytic activity of AlpH to various extent (Fig. 6e, Supplementary Fig. 16). Interestingly, the K266D mutant almost completely lost its catalytic activity. Double mutations of H270A/D271A and D271A/H267A further decreased the AlpH activity to around 5% of the wild-type AlpH. These experimental results verified the importance of the residues identified from the docking analysis.

## Discussion

Diazo NPs are hot spots in NP biosynthesis studies due to their structural rarity and broad spectrum of bioactivities[37]. Our understanding of the biosynthesis of the diazo NPs has been significantly improved owing to recent biochemical studies in cremeomycin[14], alazopeptin[4], and others[1,6,38]. Specifically, the secondary metabolism-specific nitrous acid biosynthetic pathway was identified to contribute a nitrogen atom for diazo group formation, which is catalyzed by an ATP-dependent ligase-CreM in cremeomycin biosynthesis and a transmembrane protein-AzpL in alazopeptin biosynthesis. Because of the identification of stealthin C from the mutant strains of the kinamycin producer, the diazo group formation in kinamycin has long been considered as a stepwise installation of the two nitrogen atoms[13]. This hypothesis is challenged by more recent studies; instead, it is speculated that the N−N bond is pre-formed in gluN$_2$H$_3$ and then incorporated into the polyketide scaffold. In kinamycin biosynthesis, the two FAD-dependent monooxygenases AlpJ and AlpK have been reported to catalyze the ring contraction reaction to generate hydroquinone-kinobscurinone. The hydroquinone-kinobscurinone was believed to be decorated by gluN$_2$H$_3$ or hydrazine to introduce the N−N bond into to polyketide scaffold. However, the critical step for the transfer of the N−N bond onto the polyketide scaffold is still missing. Herein, we identified an OMT enzyme, AlpH, which catalyzes a SAM-independent installation of gluN$_2$H$_3$ onto the polyketide scaffold.

Our biochemical data suggested a Mannich reaction-mediated gluN$_2$H$_3$ transfer and five-membered ring closure in the kinamycin biosynthesis. The putative key aldehyde/acid intermediate **7**, generated from DHR by AlpJ and AlpK, condenses with gluN$_2$H$_3$ to form a hydrazone intermediate **8**, which subsequently undergoes an enolate anion attack to the iminium to close the five-membered ring, resulting in intermediate **9** (Fig. 4a). Therefore, kinobscurinone, a long believed kinamycin biosynthetic intermediate, might be an artifact. The intermediacy of kinobscurinone is mainly proposed from early feeding experiments, although with a very low incorporation rate[13]. AlpK is proposed to hydroxylate the benzofluorene intermediate to give hydroquinone-kinobscurinone due to its importance in seongomycin production[22]. However, no other direct biochemical evidence for this hydroxylation step was provided. Recently, Wang et al. reported that seongomycin could also be produced without AlpK, which is in contradiction to previous results[23]. After the ring closure, **9** is oxidized to

form a relatively stable intermediate, **5**. The identification of **5** clearly demonstrated that gluN$_2$H$_3$ is directly transferred to the polyketide scaffold and ruled out the possibility of the hydrazine transfer suggested in previous studies. **5** could be gradually converted to pre-kinamycin, confirming it as a genuine intermediate in kinamycin biosynthesis. Mannich reactions have mainly been implicated in alkaloid NPs biosynthesis and are rarely described in other NPs biosynthesis[39]. Also, there are only limited examples of enzymes that can catalyze Mannich reactions. To our knowledge, LkcE is the only enzyme (except for promiscuous enzymes) to catalyze an enzymatic intramolecular Mannich reaction in LC-KA05 biosynthesis[40]. In light of the data in this study, AlpH represents the second enzyme catalyzing a Mannich reaction in the biosynthesis of aromatic polyketide kinamycin. Considering the docking and site-directed mutation results, we propose a mechanism for the AlpH catalyzed reaction (Supplementary Fig. 25). The Mannich reaction may be initiated through the protonation of the carbonyl group of the aldehyde intermediate (**7**) by a general acid (H267) to generate an iminium intermediate. Subsequently, the enol/enolate attack the iminium followed by decarboxylation to form the five-membered ring (Supplementary Fig. 25).

SAM-dependent methyltransferase family proteins from various NPs BGCs mainly function as methyltransferases that enriched their structures[41]. Intriguingly, there is a group of SAM-dependent methyltransferase family enzymes catalyzing diverse non-methylation reactions such as hydroxylation, decarboxylation, cyclization, and others[42]. Although they catalyze methylation-unrelated reactions, SAM is usually a required cofactor, such as SlnM[43] in salinomycin biosynthesis and LepI[44] in leporin C biosynthesis. AlpH is evolutionarily related to a family of small molecule OMT as indicated by their high amino acid sequence similarity. HHpred analysis also indicated the most similar structure of AlpH is OMT-LaPhzM[34] which agrees with an overall similar crystal structure. However, AlpH doesn't catalyze a methyl transfer reaction and purified AlpH is free of SAM. Biochemical assays and mutagenesis experiments demonstrated that AlpH catalyzes a SAM-independent Mannich reaction including hydrazine-carbonyl condensation and the following ring closure. In the AlpH-catalyzed reaction, the two small molecule substrates, gluN$_2$H$_3$ and an aromatic polyketide intermediate **7**, may need a relatively large active site. We propose that an empty pseudo-SAM binding site may be suitable for accommodating gluN$_2$H$_3$ and the putative aromatic polyketide intermediate **7**. The crystal structure of AlpH together with our docking analyses is consistent with this hypothesis. The condensation product **9** could be properly docked into the active site, suggesting the pseudo-SAM binding site may be suitable for gluN$_2$H$_3$ binding. Surface plasmon resonance analysis of AlpH and gluN$_2$H$_3$ also identified a specific interaction between them (Supplementary Fig. 26). Recently, two homologous groups of periclases were characterized as SAM-independent OMT-like enzymes[35]. Our phylogenetic analysis revealed that AlpH and its homologs form a separate clade from these periclases and their evolution distance is not close compared with other OMT family enzymes (Supplementary Fig. 17). Therefore, AlpH may represent a new functional subtype of OMT enzymes.

Cofactors are key components of many enzymes and possess extensive catalytic potential for chemical transformations[45,46]. Many enzymes gain new functions through binding of new cofactors[47]. Our results suggest an interesting evolution process for AlpH. The protein shares a very similar 3D structure to the OMT family proteins. Intriguingly, the loss of SAM binding is critical for AlpH, providing an additional binding site for gluN$_2$H$_3$. During the evolution of AlpH, the putative OMT ancestral protein seems to undergo a series of mutations to decrease its binding affinity to SAM but increase its binding affinity to gluN$_2$H$_3$. Overall, we described the preformed N−N bond incorporation onto the polyketide scaffold and completed the long missing steps for diazo installation in kinamycin biosynthesis. The recruitment of an OMT-like enzyme, AlpH, for gluN$_2$H$_3$ transfer may

provide a new strategy of protein engineering for novel functions in the future.

## Methods

Bacterial strains, plasmids, and BACs used in this study are listed in Supplementary table 1, and PCR primers are listed in Supplementary Table 2. General enzymes, chemicals, Kits, media, and molecular biological reagents were purchased from standard commercial sources. The *E. coli* BAP1[48] was fermented in a production medium at 18 °C. The *E. coli* strains were cultured in LB at 37 °C or 30 °C supplemented with appropriate antibiotics as needed. *E. coli* DH5α was used for plasmid propagation, BL21(DE3) for protein expression, and *E. coli* BW25113/pIJ790[49] was used for PCR-targeted mutagenesis. *E. coli* ET12567 bearing the RK2-derived helper plasmid pUB307[50] was used to facilitate the intergeneric triparental conjugation from *E.coli* ET12567/BACs to *S. albus* J1074[51] and their derivatives were grown on soya flour medium agar for sporulation and conjugation, in R2 liquid medium containing 5% HP-20 resin for fermentation.

### Sequence alignment and phylogenetic analysis

The phylogenetic tree was constructed by MEGA7[52] using a neighbor joining method. Multiple-sequence alignment was constructed by ClustalW [53] and ESPrit 3.0[54]. Protein sequences used for bioinformatic analysis were listed in the Supplementary table 4.

### Heterologous expression of BACs

The transfer of 3C2 and its derivatives from *E. coli* to *S. albus* J1074 was accomplished using *E. coli* ET12567/pUB307-mediated triparental conjugation. Integration of the BACs was confirmed by apramycin resistance and diagnostic PCR.

### Production and analysis of kinamycin and related metabolites

For the productions of kinamycin and related metabolites, *Streptomyces* spore suspensions were inoculated in liquid trypticase soy broth (TSB) and incubated at 30 °C for 24–48 h as seed culture. A total of 1 mL of seed culture was then transferred into 35 mL of R2 fermentation medium and incubated with shaking (220 rpm) for an additional 2-3 days at 30 °C.

For the reconstitution of diazo formation, *E. coli* BAP1/pGro7/pXY-2/pXY-3/pXY-6 was used as the starting strain for the strain construction of this work[31]. The gene *alpJ* and *alpK* fragments excised by *Xba* I and *Hind* III were sequentially ligated into the *Spe* I and *Hind* III sites of the pXY-2 just behind MCAT (malonyl-CoA: acyl carrier protein transacylase) to form pXY-2JK, the gene *alpH* was inserted into pXY-6 behind *alpG* to form pXY-6H. The BAP1 strains Bjk and Bjkh were inoculated into 3 mL LB liquid medium with required antibiotics and cultured overnight at 37 °C with shaking at 220 rpm. Then 500 μL of overnight culture was transferred to 50 mL fresh production medium with required antibiotics and cultured at 37 °C until the OD$_{600}$ reached between 0.4 and 0.6. IPTG (isopropyl-β-d-thiogalactopyranoside) (final concentration 500 μM) and arabinose (final concentration 3 mM) were added and the cultures were grown at 18 °C for an additional 2–3 days.

For metabolite analysis, the culture broth was extracted with an equal volume of ethyl acetate after adding 1% glacial acetic acid. The organic extract was dried by rotary evaporation, and the dried material was dissolved in methanol for subsequent analysis. HPLC analysis was performed on an Agilent 1200 HPLC system using an Eclipse XDB-C18 column (5 μm, 4.6 × 150 mm). In each experiment, the following condition was used: 75% solvent A (water with 0.1% trifluoroacetic acid [TFA]) to 100% solvent B over 20 min and 100% solvent B for 10 min at a flow rate of 1 mL/min.

### Isolation of DHR and prekinamycin

For the preparation of DHR, a 3 L liquid fermentation of the *E. coli* BAP1/pGro7/pKM-1/pXY-3/pXY-6 was extracted with an equal volume of ethyl acetate after adding 1% glacial acetic acid[55]. The ethyl acetate was removed by vacuum evaporation. The ethyl acetate extract was loaded onto the silica gel column and flushed with chloroform. DHR was eluted at 100% chloroform. The fractions containing DHR were collected and pooled, and further purified by reverse-phase semi-preparative HPLC. At last, 1.1 g of DHR was obtained. For preparation of prekinamycin, a 5 L liquid fermentation broth of the mutant Δ*alp1S-1V* was extracted by ethyl acetate after adding 1% glacial acetic acid. The ethyl acetate was removed by vacuum evaporation, and the extract was dissolved in methanol, and then applied to reverse phase C18 column chromatography, eluted with a gradient elution of H$_2$O: MeOH mixture. The fractions were further purified by semi-preparative HPLC to afford pure prekinamycin.

### Chemical synthesis of L-glutamylhydrazine

16 mL hydrazine hydrate (90%) and 8 g L-glutamic acid-5-methyl ester were mixed and stirred for 5–7 h at 30 °C. Methanol was then added and the reaction mixture was kept at 4 °C overnight. The solids were filtered out and washed with methanol and dried to obtain 6 g gluN$_2$H$_3$. The yield of L-glutamylhydrazine was 75%. In the end, 2.0 mg of gluN$_2$H$_3$ was dissolved in D$_2$O for NMR experiments.

### Feeding experiments in *E. coli* BAP1

The BAP1 strains Bjk and Bjkh were inoculated into 3 mL of LB liquid medium with required antibiotics and cultured overnight at 37 °C with shaking at 220 rpm. Then 500 μL of overnight culture was transferred to 50 mL fresh production medium with required antibiotics and cultured at 37 °C. When OD$_{600}$ reached 0.4–0.6, IPTG (500 μM) and arabinose (3 mM) were added. After growing at 18 °C for one day, gluN$_2$H$_3$ was fed into the culture at a final concentration of 1 mM. The culture was grown for another two days at 18 °C.

### Protein expression and purification

The *alpH*, *alpJ*, and *alpK* genes were PCR amplified using 3C2 as a template and cloned into pET28a vector individually using appropriate restriction sites (seen in Supplementary table 2) and then were transformed to DH5α. The error-free plasmids were verified by sequencing and then transformed into *E. coli* BL21(DE3) for protein expression. Transformants were cultivated overnight at 37 °C in LB with 100 μg/mL kanamycin, and 10 mL of the overnight culture was then used to inoculate 1 L of LB with 100 μg/mL kanamycin. Cultivation continued at 37 °C with shaking at 220 rpm, until the OD$_{600}$ reached between 0.4 and 0.6, and IPTG was added to 500 μM. The cultures continued to grow at 16 °C for an additional 18 h. The *alp1W* gene was amplified from 3C2 and cloned into pWY45 using *Nde* I and *EcoR* I restriction sites to generate pWY45-*alp1W*. The error-free plasmids were verified by sequencing and introduced into *S. lividans* SBT5 and plated on MS-agar medium containing apramycin (50 μg/mL) and trimethoprim (50 μg/mL) for 4 days at 30 °C. Positive exconjugants were cultivated for 3 days at 30 °C in 50 mL TSBY with apramycin (50 μg/mL) and trimethoprim (50 μg/mL), and 15 mL of the culture was then used to inoculate 250 ml YEME medium with apramycin (50 μg/mL) for 3 days at 30 °C. Then thiostrepton (25 μg/mL) was added to induce protein expression for another 24 h. For protein purification, cells were harvested by centrifugation (8,000 g) at 4 °C and re-suspended in the binding buffer (20 mM Tris-HCl, 500 mM NaCl, 5 mM imidazole, pH 8.0). Then cells were disrupted by sonication. Following centrifugation, the supernatant was loaded onto the prewashed Ni-NTA column. The crude proteins were washed with the washing buffer (20 mM Tris-HCl, 500 mM NaCl, 30 mM imidazole, pH 8.0) and then eluted with elution buffer (20 mM Tris-HCl, 500 mM NaCl, 500 mM imidazole, pH 8.0). The eluate was then concentrated to a volume of 2.5 mL using an Amicon spin filter with a 10 kDa molecular weight cutoff. Excess imidazole was removed using a PD-10 column (GE Healthcare) equilibrated using storage buffer (50 mM Tris-HCl, 300 mM NaCl, 10%

glycerol). Protein concentrations were determined by Bradford method and the protein purity was assessed by SDS−PAGE.

## In vitro enzyme reactions

In AlpH catalyzed reaction, 0.25 mM gluN$_2$H$_3$ (1 mM L-Cys or 1 mM NAC was used when necessary), 1 mM TCEP, 1 mM SAM, 2 mM NADH, 5 μM FAD, 40 μM AlpJ, 25 μM AlpK, 35 μM AlpH, and 120 μM DHR were incubated in pH 7.5 50 mM Tris-HCl buffer at 30 °C for 35 min. In the in vitro assay of Alp1W, 4.5 μM Alp1W was used to replace AlpH. At the end of the reaction, an equal volume of ethyl acetate with 1% glacial acetic acid was used to extract the product twice. The organic extract was dried by rotary evaporation and the final extract was dissolved in 50 μL DMSO. For direct analysis of compound **5** in the reaction mixture, half a volume of cold methanol was added to the reaction mixture at the end of the reaction and the supernatant was directly analyzed by HPLC after centrifugation. To prepare the standard seongomycin, AlpJ, AlpK, NADH, FAD, NAC, and DHR were incubated together in 50 mM Tris-HCl buffer (pH 7.5) at 30 °C for 2 h and quenched by adding 400 μL cold methanol. The extract was dried by rotary evaporation, and the final extract was dissolved in 50 μL methanol. To prepare the standard stealthin C, AlpJ, AlpK, NADH, FAD, L-Cys, and DHR were incubated in 50 mM MOPS buffer (pH7.5) at room temperature for 2 h and quenched by adding 400 μL cold methanol. The extract was dried by rotary evaporation, and the final extract was dissolved in 50 μL methanol.

## Construction of gene disruption mutants and complementation mutants

All genetic manipulations on BAC 3C2 were conducted using λ-Red recombination-mediated PCR-targeted gene deletion[56]. Individual gene disruption cassettes were generated via PCR amplification of the kanamycin resistance gene cassette with flanking FRT sites from pJTU4659. Primers were designed with 39 nucleotides matching the adjacent sequences of the targeted gene. Gene replacement by λ-Red recombination was achieved in *E. coli* BW25113/pIJ790/3C2. Mutated constructs were confirmed by PCR and then transferred to *E. coli* BT340 for flippase (FLP) recombinase-mediated excision of the kanamycin resistance gene. The corresponding final knockout BAC, with FLP scar, was verified by PCR before transfer into *S. albus* J1074. For complementation experiments, the integrative plasmid pPM927 was used. The *alpH* gene was amplified from BAC 3C2 using primers AlpH-*Nde* I-F and AlpH-*EcoR* I-R and was ligated to the pre-digested pJTU968 to generate the construct pJTU968-*alpH*. The fragment of *alpH* under the control of *kasO*p* promoter was cut from pJTU968-*alpH* by digestion with *Mun* I and *EcoR* I and was then ligated to *EcoR*I single-digested and dephosphorylated plasmid pPM927 to generate plasmid pPM927-*alpH*. The resulting pPM927-*alpH* was introduced into the *S. albus* J1074/W-3C2Δ*alpH* mutant to obtain the complementation strain Δ*alpH*:: *alpH*. Exconjugants were selected on MS agar plates with thiostrepton, apramycin, and trimethoprim. The *alp1W* gene was amplified from BAC 3C2 using primers Alp1W-onestep-F and Alp1W-onestep-R and was cloned into pPM927. The resulting pPM927-*alp1W* was introduced into the *S. albus* J1074/W-3C2Δ*alp1W* mutant to obtain the complementation strain Δ*alp1W*:: *alp1W*.

## Measurement of the presence of SAM in AlpH

To detect whether SAM was present in AlpH, 360 μM AlpH in 40 μL storage buffer (50 mM Tris-HCl, 300 mM NaCl, 10% glycerol) was denatured by acetonitrile. 100 μM SAM was treated by acetonitrile as a control. Then the solutions were centrifuged, and the supernatants were analyzed by LC-MS. The standard of SAM was also analyzed by LC-MS.

## Protein crystallization and structural elucidation

Crystals of AlpH were obtained by the hanging drop vapor diffusion technique at 16 °C. Freshly purified AlpH protein (10 mg/mL or

20 mg/mL in 20 mM Tris-HCl, pH 7.5, 100 mM NaCl, 1 mM DTT, and 1 mM EDTA) was mixed with equal volumes of reservoir solution containing 0.03 M citric acid, 0.07 M Bis-Tris propane (pH 7.6), and 20% (w/v) PEG3350. Crystals were frozen using reservoir solution added 20% (v/v) glycerol as cryo-protectant. A 1.87 Å resolution X-ray data set was collected at the beamline BL10U2 of the Shanghai Synchrotron Radiation Facility (SSRF)[57]. The diffraction data was processed and scaled with XDS and autoPROC software suite[58,59] (Supplementary table 5).

The phase problem of the complex was solved by the molecular replacement method using the automatic molecular replacement pipeline MoRDA within the CCP4 suite[60,61]. The initial model was rebuilt manually using COOT[62], and then refined using REFMAC[63], and PHENIX[64]. The qualities of the final model were validated by MolProbity[65]. The final refinement statistics of solved structure in this study were listed in Supplementary table 5. All the structural diagrams were prepared using the program PyMOL (http://www.pymol.org/).

## Molecular docking of product to AlpH

Rigid molecular docking was performed using AutoDock Vina 1.2.0[66]. The ligand product **9** was generated using ChemBio3D Ultra 12.0, and then energy minimized with Molecular Mechanics (MM2) until a minimum root-mean-square (RMS) gradient of 0.100 was achieved (https://www.chemdraw.com.cn/). Water molecules were removed from the AlpH structure in preparation for docking. The Gasteiger charges and hydrogens were added to the receptor structure and ligand with AutoDocktools[67]. When docking, the explorable space for docking was defined as a cube 20 Å in length centered at −67.713, 18.156, and −36.454 including the entire active-site, other key parameters such as algorithm were set to default values. Finally, the complex structure with the best combination of low binding energy and favorable orientation was selected. PyMOL 2.5 (http://www.pymol.org) was used for viewing the molecular interactions and image processing.

## Measurement of the binding affinity between gluN$_2$H$_3$ and AlpH

The binding affinity was determined by surface plasmon resonance (SPR) using the Biacore 8 K. AlpH was diluted to 20 μg/ml in pH 4.5 sodium acetate buffer and immobilized onto a CM5 sensor chip. GluN$_2$H$_3$ was diluted into 0.0975, 0.195, 0.39, 0.78, and 1.56 μM using PBS-0.05% Tween20 buffer. The experiment was performed in PBS-0.05% Tween20 buffer at a flow rate of 30 μl/min for 120 s (contact time) and another 120 s (dissociation time). The $K_{on}$, $K_{off}$, and $K_d$ values were determined using Biacore™ Insight Evaluation Software version 4.0 with a 1:1 binding model.

## Molecular dynamics (MD) simulations

Three independent 200 ns MD simulations for ligand-bound systems were performed by using Gromacs 2018.3[68]. The a99SB-ILDN force field[69] was utilized for the protein, while the TIP3P model[70] was employed for water molecules. For the ligand, Ambertools 20 antechamber[71] was used to process its topology using GAFF force field[72] and AM1-BCC charge type[73]. The net charge of the ligand was determined to be 0. The system was solvated in a dodecahedron box with dimension of 11.996 × 11.996 × 11.996 nm$^3$, and it was filled with 36,212 water molecules. Na$^+$ and Cl$^-$ were added to obtain a salinity of 0.15 M and maintain neutral total charge, resulting in a total of 119753 atoms in the system.

Prior to commencing the production simulations, an energy minimization step was performed using the steepest descent method. The maximum force was set to 1000 kJ mol$^{-1}$ nm$^{-1}$, and convergence was achieved within 50,000 steps. Subsequently, a 100 ps NVT equilibrium simulation was conducted, applying a position restraint on heavy atoms with a force constant of 1000 kJ mol$^{-1}$ nm$^{-2}$. The LINCS method was used to constrain all covalent bonds, allowing a time step of 2 fs[74]. The van der Waals interaction cutoff was also set to 1.0 nm.

Long-range electrostatic interactions were calculated using the Particle Mesh Ewald method with a cutoff of 1.0 nm[75].

Temperature control was maintained around 310 K using the V-rescale coupling method[76]. Following the NVT equilibrium simulation, a 100 ps NPT equilibrium simulation was performed, still retaining the position restraints and using the Parrinello Rahman method[77] for pressure coupling. Finally, 200 ns NPT production simulations with no restraints were carried out, the remaining parameters remained consistent with the NPT equilibrium simulations.

### Reporting summary

Further information on research design is available in the Nature Portfolio Reporting Summary linked to this article.

## Data availability

The coordinate and structure factor of AlpH solved in this study have been deposited in the Protein Data Bank under the accession code 8H3T. All other relevant data supporting the findings of this study are available in this manuscript and Supplementary Information. Source data are provided with this paper.

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

## Acknowledgements

This work was supported by the National Key Research and Development Program of China (2021YFC2100600 and 2018YFA0900402), the National Science Foundation of China (32170076, 31870026, and 21621002), the Science and Technology Commission of Shanghai Municipality (20XD1425200) (to L.P.), and the open project of the State Key Laboratory of Bioorganic and Natural Products Chemistry, Shanghai Institute of Organic Chemistry, Chinese Academy of Sciences. We thank SSRF BL10U2 for X-ray beam time. We thank Prof. Min Xu of Tianjin Institute of Industrial Biotechnology, Chinese Academy of Sciences and Prof. Keqiang Fan of the Institute of Microbiology, Chinese Academy of Sciences for helpful discussion.

## Author contributions

Y.C.Z., X.Y.L., L.J.H., Y.W., S.J.L., P.S., and X.Z.W. performed compound isolation and structure determination. Y.C.Z., X.Y.L., and J.Z. conducted the in vitro biochemical studies and analysis. Y.C.Z. and L.F.P carried out the protein crystallization experiments, X-ray analysis, and the crystal structure determinations of the proteins. Z.H.X performed molecular docking analysis. X.Y.S. and W.N.W conducted the molecular dynamics simulations. Y.C.Z., X.Y.L., X.Y.H., and M.J. analyzed the data and wrote the manuscript. M.J. and Z.X.D directed the research. Y.C.Z and X.Y.L contributed equally to this work.

## Competing interests

The authors declare no competing interests.
