## [Peer review file · Nature Communications]

REVIEWER COMMENTS

Reviewer #1 (Remarks to the Author):

In this paper, the authors have successfully identified the key enzyme for diazo group formation in the biosynthesis of kinamycin. First, the authors performed gene disruption using a heterologous expression system and discovered AlpH, a candidate key enzyme that catalyzes diazo group synthesis. AlpH was expected to be a SAM-dependent methyltransferase from its amino acid sequence. Therefore, it was unexpected that such an enzyme would be involved in the diazo group synthesis. Next, the authors carried out in vitro analysis using a recombinant protein of AlpH. Because of the difficulty of preparing the substrate of AlpH, they have developed a method to analyze the function of AlpH by using a sequential reaction using three enzymes, AlpK, AlpJ and AlpH, from DHR. As a result, the authors succeeded in the in vitro synthesis of prekinamycin, which has a diazo group, from DHR and glutamylhydrazine. Furthermore, they have succeeded in detecting an N-acylhydrazine intermediate (9) as a precursor of prekinamycin by changing the reaction conditions. Prekinamycin seemed to be synthesized by non-enzymatic oxidation and the amide cleavage of this intermediate (9). Furthermore, by examining the specificity of the formation of compound 2 and compound 5 (analog of 9) under various conditions, they predict that the direct substrate of AlpH is an aldehyde (7) which is synthesized prior to the formation of the five-membered ring of benzofluorene (6). Next, the crystal structure of AlpH was solved. Based on the structure obtained, it was predicted that AlpH lacks the ability to bind SAM. Unfortunately, the structure of the substrate-enzyme complex has not been solved, but Docking analysis predicted that the substrate would bind to the cavity, which corresponds to the SAM-binding site. The authors performed site-directed mutagenesis of the predicted substrate-binding site. The results showed that a single amino acid substitution resulted in decreased activity. The identification of AlpH is an interesting study that identifies a mechanism of diazo group synthesis in kinamycin and shows that methyltransferases can evolve into enzymes that catalyze characteristic reactions in a SAM-independent manner. Most experiments were performed clearly and the results provide strong evidence that the AlpH is responsible for the hydrazino moiety incorporation into the polyketide scaffold. The weak point is that the actual substrate of AlpH is not clearly identified and the discussion on the reaction mechanism of AlpH is not enough.

Please describe the function of AlpJ and AlpK in more detail in line 70. Since the substrate of AlpH is the product of AlpJ and/or AlpK, the actual function of these enzymes should be important. It is not clear what kind of reaction AlpJ and AlpK catalyze to make the actual substrate of AlpH in the current state. Therefore, the function of these enzymes should be carefully described (or the authors can perform additional experiment to get information about their function). Is there any possibility that the intermediate other than 7 generated during the AlpJ and AlpK reactions might be the substrate of AlpH? The reaction containing only AlpJ or AlpK, and AlpH with AlpJ or AlpK should also be examined. In these reactions, Fre can also be included, as reported in reference 20. In reference 20, benzofluorene (6) and its dimer seemed to be synthesized by incubation of DHR with only AlpJ with Fre.

In figure 4a, 8 to 9 seemed to also require dehydrogenation. Please confirm.

The single amino acid substitution of the AlpH in Figure 6e seemed to only show a moderate effect on the activity of AlpH. Authors can try additional mutagenesis experiments (inverting the physicochemical property of side chains). For instance, K266 seemed to be interacting with the carboxylic acid of glutamylhydrazine. Substitution of K266 to acidic amino acids may decrease the activity drastically.

Because authors are proposing that AlpJ catalyzes C-C bond formation in pathway 2 of Figure 4a, is there any possibility that the AlpJ also catalyzes C-C bond formation in pathway 1 of Figure 4a? I recommend authors carry out docking simulation 8 considering this possibility.

Can authors propose more detailed reaction mechanism of C-N bond formation and C-C bond formation presumably catalyzed by AlpH? Are there any amino acid residues that might function as catalytic acid (or base) near the active site?

If it is possible, the reliability of the docking model should also be confirmed by the MD simulation.

Line 121, please add a reference here.

Line 262, the following reference should be cited here.

<https://pubs.acs.org/doi/10.1021/ja031724o>

The biosynthesis of jadomycin B aglycon and prekinamycin is similar at the beginning but not at the later stage because C-C bond is synthesized in prekinamycin biosynthesis while C-N bond is synthesized in jadomycin B biosynthesis. This should be considered. Is there any possibility that supplementation of isoleucine to the reaction mixture or heterologous expression strain without AlpH results in jadomycin B-like compounds? If such a reaction can proceed, does it compete with prekinamycin synthesis? The such experiment should strengthen the hypothesis in which the 7 is the substrate of AlpH proposed by the authors.

Line, 298-300, please describe RMSD values to the corresponding structures

Figure 4, does hydroquinone-kinobscurinone lack one double bond?

Please provide the SDS-PAGE of all AlpH variants used in this study.

Figure S13, please provide n = 3 data. How about the influence of other compounds such as SAH and sinefungin.

Table S6, S7, please add the position number.

Line 213, please provide the UV spectra of compound 5

Figure S11, (b) which kind of chromatograms are shown here?

Reviewer #2 (Remarks to the Author):

Zhao et al. have discovered an unprecedented SAM-independent methyltransferase like enzyme AlpH that installs the glutamylhydrazine (gluN2H3) molecule onto the kinamycin scaffold through a unique Mannich reaction manner. This work has challenged traditional views about this biosynthetic pathway, including novel discoveries such as 1) gluN2H3 but not hydrazine is installed; 2) AlpH but not previously proposed amidotransferase Alp1W catalyzes the installation; 3) gluN2H3 is installed before 5-membered B-ring forms, and thus several intermediates found or proposed previously with the contracted B-ring are not on-pathway. All of these findings will significantly contribute to the understanding of this pathway to generate the diazo containing kinamycin and other related natural products. Although the work is significant and interesting, more data are needed to further support the conclusions and scholarly presentation shall also be improved before publication. Here are some specific comments:

1. In Fig. 2a, the retention times of compound 3 stealthin C and kinamycin D are too close to each other, which makes the data analysis a little difficult. There seems to be kinamycin D formation but no 3 in Δ alp2F2G strain (rxn iv). The authors should explain why (endogenous HNO₂ provider?).
2. In Line 185, the authors mentioned the formation of new product 4 but not clearly labeled in Fig. 2b. Even if it has been confirmed to be the same as prekinamycin, the authors should provide a label such as "4 = prekinamycin" in the figure. Identical UV absorbance spectrum has been mentioned which should be shown in SI.

3. The authors should provide more Ctrl for the one-pot reaction shown in Fig. 3a. There are multiple enzymes (AlpJ/K/H) and cofactors (FAD, NADH) in the system. Please provide HPLC profiles for Ctrl reactions eliminating each component in SI.
4. Since there is time-course assay presenting in Fig. 3b, the authors should also mention the incubation time for the one-pot reactions shown in Fig. 3a. There are different amounts of compound 1 and 6 in “-gluN2H3” and “Boiled AlpH” traces shown in 3a and 3b. Is this caused by different incubation time, or different sample treatment methods (solvent extraction vs. direct injection)?
5. There is no need to refer to compound 5 (fig 3b) as X. Remove this X label.
6. There are many mistakes in Fig. 4a. The structures of prekinamycin, hydroquinone-kinobscurinone, kinobscurinone are not consistent with Fig 1. Please double check the mechanisms carefully.
7. Please further clarify the stepwise assays listed in Fig. 4c. The current labels are very confusing.
8. Line 375, the physiological function of AlpK remains to be elucidated? This is confusing as all in vitro assays in this manuscript use AlpK.
9. Please further proofread the entire manuscript to correct grammar errors. In addition, more refs are needed. For example, line 408.

Reviewer #3 (Remarks to the Author):

Diazo are promising natural compounds. In this manuscript, the authors focused on the biosynthesis of kinamycin and the mechanism for the installation of the hydrazine group onto the kinamycin scaffold. They discovered an unprecedented O-methyltransferase-like protein, AlpH, responsible for the hydrazine incorporation in kinamycin biosynthesis. Biochemical and structural characterizations of AlpH have been performed and support the critical role of AlpH in a diazo NP biosynthesis. This manuscript brings novel insights in the biosynthesis of diazo compounds and regarding SAM-independent OMT-like enzyme. I recommend publication after major revision.

Although AlpW is required for kanamycin biosynthesis, its role is unclear. Is AlpW conserved in biosynthesis gene cluster? What is the assigned function of AlpW in database? This should be discussed. In the Δ alpW strain, complementation experiment should be performed to support the critical role of AlpW.

HRMS analysis should be provided for all products shown in HPLC analysis to confirm their identity.

It should be discussed in which biosynthesis pathways are found AlpH homologues (Fig S16) and their protein sequence/structure/active site homology.

The docking model is interesting and provides a rational for SAM-independent activity of AlpH. However, the structure of AlpH with the substrate (gluN2H3 and/or a polyketide intermediate) would further support the role of AlpH. Substrate-enzyme structure and/or binding affinity experiment should be performed.

Complete listing of the reviewer comments and our responses

- Reviewers' comments, black.
- Our responses, blue.

Reviewer #1 (Remarks to the Author):

In this paper, the authors have successfully identified the key enzyme for diazo group formation in the biosynthesis of kinamycin. First, the authors performed gene disruption using a heterologous expression system and discovered AlpH, a candidate key enzyme that catalyzes diazo group synthesis. AlpH was expected to be a SAM-dependent methyltransferase from its amino acid sequence. Therefore, it was unexpected that such an enzyme would be involved in the diazo group synthesis. Next, the authors carried out in vitro analysis using a recombinant protein of AlpH. Because of the difficulty of preparing the substrate of AlpH, they have developed a method to analyze the function of AlpH by using a sequential reaction using three enzymes, AlpK, AlpJ and AlpH, from DHR. As a result, the authors succeeded in the in vitro synthesis of prekinamycin, which has a diazo group, from DHR and glutamylhydrazine. Furthermore, they have succeeded in detecting an N-acylhydrazine intermediate (9) as a precursor of prekinamycin by changing the reaction conditions. Prekinamycin seemed to be synthesized by non-enzymatic oxidation and the amide cleavage of this intermediate (9). Furthermore, by examining the specificity of the formation of compound 2 and compound 5 (analog of 9) under various conditions, they predict that the direct substrate of AlpH is an aldehyde (7) which is synthesized prior to the formation of the five-membered ring of benzofluorene (6). Next, the crystal structure of AlpH was solved. Based on the structure obtained, it was predicted that AlpH lacks the ability to bind SAM. Unfortunately, the structure of the substrate-enzyme complex has not been solved, but Docking analysis predicted that the substrate would bind to the cavity, which corresponds to the SAM-binding site. The authors performed site-directed mutagenesis of the predicted substrate-binding site. The results showed that a single amino acid substitution resulted in decreased activity. The identification of AlpH is an interesting study that identifies a mechanism of diazo group synthesis in kinamycin and shows that methyltransferases can evolve into enzymes that catalyze characteristic reactions in a SAM-independent manner. Most experiments were performed clearly and the results provide strong evidence that the AlpH is responsible for the hydrazino moiety incorporation into the polyketide scaffold. The weak point is that the actual substrate of AlpH is not clearly identified and the discussion on the reaction mechanism of AlpH is not enough.

Please describe the function of AlpJ and AlpK in more detail in line 70. Since the substrate of AlpH is the product of AlpJ and/or AlpK, the actual function of these enzymes should be important. It is not clear what kind of reaction AlpJ and AlpK

catalyze to make the actual substrate of AlpH in the current state. Therefore, the function of these enzymes should be carefully described (or the authors can perform additional experiment to get information about their function). Is there any possibility that the intermediate other than 7 generated during the AlpJ and AlpK reactions might be the substrate of AlpH? The reaction containing only AlpJ or AlpK, and AlpH with AlpJ or AlpK should also be examined. In these reactions, Fre can also be included, as reported in reference 20. In reference 20, benzofluorene (6) and its dimer seemed to be synthesized by incubation of DHR with only AlpJ with Fre.

We thank the reviewer for the important suggestions. The functions of AlpJ and AlpK were further described in the introduction of the revised manuscript. As the direct substrate of AlpH could not be purified and characterized because of its instability in this work, we proposed a Mannich reaction catalyzed by AlpH using 7 as a potential substrate. We agree that it is possible that other intermediate, such as kinobscurinone, might be the substrate of AlpH. However, according to our competition experiments, we believe that the aldehyde intermediate, 7, is the more reasonable substrate for AlpH. The various control experiments containing only AlpJ or AlpK, and AlpH with AlpJ or AlpK have done as suggested (Fig. S12). Also, Fre was included in various control experiments.

Fig. S12. HPLC analysis of the reactions with different combinations of enzymes and cofactors for the one-pot reaction.

In figure 4a, 8 to 9 seemed to also require dehydrogenation. Please confirm.

We appreciate the reviewer pointing out our mistake. The figure has been revised.

Fig. 4a. Mechanistic alternatives for the coupling of gluN_2H_3 and polyketide intermediate.

The single amino acid substitution of the AlpH in Figure 6e seemed to only show a moderate effect on the activity of AlpH. Authors can try additional mutagenesis experiments (inverting the physicochemical property of side chains). For instance, K266 seemed to be interacting with the carboxylic acid of glutamylhydrazine. Substitution of K266 to acidic amino acids may decrease the activity drastically.

We appreciate the important suggestion. Additional mutagenesis experiments have been performed as suggested (Fig. 6e, S16).

Fig. 6e. Relative activities of AlpH mutants.

Fig. S16. HPLC profiles of the reactions catalyzed by wild-type AlpH and its mutants.

Because authors are proposing that AlpJ catalyzes C-C bond formation in pathway 2 of Figure 4a, is there any possibility that the AlpJ also catalyzes C-C bond formation in pathway 1 of Figure 4a? I recommend authors carry out docking simulation 8 considering this possibility.

We thank the reviewer for this important comment. It is a reasonable hypothesis that AlpJ catalyzes C-C bond formation in pathway 1 of Fig. 4a. The docking simulations were conducted as suggested and **8** could be docked into the active site of AlpJ. However, two unrelated aromatic polyketide compounds could also be docked into the active site of AlpJ (the result was shown below). We reasoned that AlpJ has a wide and open active site which is suitable for interactions with aromatic polyketide compounds. Thus, the docking results could not help us infer that AlpJ catalyzes C-C bond formation in pathway 2 of Fig. 4a. As a second step of Mannich reaction, C-C bond formation is highly favorable. Therefore, we believe that it might be more possible that the C-C bond formation happens in the active site of AlpH after the condensation of gluN_2H_3 and the aldehyde intermediate.

Figure Response 1. Surface representations and the stick-ball models showing the overall docking models of AlpJ in complex with different polyketide compounds (a) product **8**. (b) murayaquinone. (c) JX111b. (d) The chemical structures of the polyketides used in docking simulations.

Can authors propose more detailed reaction mechanism of C-N bond formation and C-C bond formation presumably catalyzed by AlpH? Are there any amino acid residues that might function as catalytic acid (or base) near the active site?

We appreciate this important suggestion. More detailed reaction mechanism of C-N bond formation and C-C bond formation has been proposed and discussed (Fig. S24). The potential catalytic base and acid are shown in the new proposed mechanism (Fig. S24).

Fig. S24. Proposed mechanism of C-N and C-C bond formation catalyzed by AlpH.

If it is possible, the reliability of the docking model should also be confirmed by the MD simulation.

Done as suggested (Fig. S22).

Fig. S22. MD simulations of the ligand-bound systems. (a) The structure of the ligand-bound state at the beginning of the MD simulation. (b) Time variation of the RMSD of the ligand molecule after superimposing the protein structures along the simulation trajectories. (c) The structure of the ligand-bound state at the end of one MD simulation trajectory. (d) The evolutions of hydrogen bond numbers related to the ligand molecule during the simulations.

Line121, please add a reference here.

Done as suggested [26].

Line 262, the following reference should be cited here.

<https://pubs.acs.org/doi/10.1021/ja031724o>

Done as suggested [33].

The biosynthesis of jadomycin B aglycon and prekinamycin is similar at the beginning but not at the later stage because C-C bond is synthesized in prekinamycin biosynthesis while C-N bond is synthesized in jadomycin B biosynthesis. This should

be considered. Is there any possibility that supplementation of isoleucine to the reaction mixture or heterologous expression strain without AlpH results in jadomycin B-like compounds? If such a reaction can proceed, does it compete with prekinamycin synthesis? The such experiment should strengthen the hypothesis in which the **7** is the substrate of AlpH proposed by the authors.

Thanks for your important suggestion. As suggested, isoleucine was added to the reaction mixture without AlpH instead of gluN₂H₃ (DHR + AlpJ + AlpK + Ile) and jadomycin A could be detected by LC-MS which is similar with previous report. This result supports the hypothesis that **7** is the substrate for AlpH. However, when isoleucine was added in the full reaction mixture (DHR + AlpJ + AlpK + AlpH + gluN₂H₃), the production of **5** was not affected. We reason that the spontaneous incorporation of isoleucine with the aldehyde intermediate (**7**) is too slow to compete with the efficient incorporation of gluN₂H₃ which is catalyzed by AlpH.

Figure Response 2. Analysis of the effect of isoleucine. (a) LC-MS analysis of DHR + AlpJ + AlpK + Ile reaction.(b) HPLC analysis of the competition reactions.

Line, 298-300, please describe RMSD values to the corresponding structures.

Correct as suggested.

Figure 4, does hydroquinone-kinobscurinone lack one double bond?

Correct as suggested.

Fig. 4a. Mechanistic alternatives for the coupling of gluN₂H₃ and polyketide intermediate.

Please provide the SDS-PAGE of all AlpH variants used in this study.

The SDS-PAGE gel of all AlpH variants used in this study has been added in supplemental information (Fig. S23).

Fig. S23. SDS-PAGE analysis of purified AlpH and its mutants.

Figure S13, please provide n = 3 data. How about the influence of other compounds such as SAH and sinefungin.

Thank you for your suggestion. All experiments have been repeated for at least three times and n=3 data were represented. The influence of SAH and sinefungin were also investigated and the results were added in Fig. S14.

Fig. S14. Effects of exogenous SAM, SAH, and Sinefungin addition to AlpJKH-catalyzed reaction.

Table S6, S7, please add the position number.

Done as suggested.

Table S7. ¹H NMR data comparison between reported prekinamycin and our isolated prekinamycin

No	This work ¹ H (mult., J in Hz)	Vladimir B. Birman et al. Report, ¹ H
1-OH	12.13 (s, 1H)	12.13(s, 1H)
9-OH	11.04 (s, 1H)	11.04(s, 1H)
3	7.79 (d, J = 7.4 Hz, 1H)	7.79 (d, J = 7.2 Hz, 1H)
2	7.61 (t, J = 7.9 Hz, 1H)	7.60 (t, J = 8.0 Hz, 1H)
4	7.24(overlapped with the CHCl ₃ peak)	7.24(overlapped with the CHCl ₃ peak)
10	6.84 (s, 1H)	6.83 (s, 1H)
12	6.71 (s, 1H)	6.71 (s, 1H)
18	2.43 (s, 3H)	2.43 (s, 3H)

Table S8. ^{13}C NMR data comparison between reported prekinamycin and our purified prekinamycin

No	This work ^{13}C	Vladimir B. Birman et al. Report, ^{13}C
16	184.8	184.9
6	181.3	181.4
1	162.4	162.5
9	154.7	154.8
11	142.0	142.0
7	136.5	136.5
3	135.9	136.0
15	133.7	133.8
5	131.8	131.9
13	128.4	128.5
2	125.3	125.4
4	120.9	121.0
8	117.7	117.8
17	117.2	115.9
10	115.8	115.5
12	114.2	114.3
14	110.4	110.5
18	29.9	29.9

Line 213, please provide the UV spectra of compound 5.
Provide as suggested (Fig. S11a).

Fig. S11a. UV absorbance spectra of compound 5

Figure S11, (b) which kind of chromatograms are shown here?

We are sorry for the confusion. Fig. S11b is the HPLC profile ($\lambda = 424$ nm) of the decomposition products of compound **X** (5).

Reviewer #2 (Remarks to the Author):

Zhao et al. have discovered an unprecedented SAM-independent methyltransferase like enzyme AlpH that installs the glutamylhydrazine (gluN2H3) molecule onto the kinamycin scaffold through a unique Mannich reaction manner. This work has challenged traditional views about this biosynthetic pathway, including novel discoveries such as 1) gluN2H3 but not hydrazine is installed; 2) AlpH but not previously proposed amidotransferase Alp1W catalyzes the installation; 3) gluN2H3 is installed before 5-membered B-ring forms, and thus several intermediates found or proposed previously with the contracted B-ring are not on-pathway. All of these findings will significantly contribute to the understanding of this pathway to generate the diazo containing kinamycin and other related natural products. Although the work is significant and interesting, more data are needed to further support the conclusions and scholarly presentation shall also be improved before publication. Here are some specific comments:

1. In Fig. 2a, the retention times of compound 3 stealthin C and kinamycin D are too close to each other, which makes the data analysis a little difficult. There seems to be kinamycin D formation but no 3 in $\Delta alp2F2G$ strain (rxn iv). The authors should explain why (endogenous HNO₂ provider?).

We are grateful for the comments. The retention times of compound 3 stealthin C and kinamycin D are very close. Thanks to their unique and different UV spectra, we could easily distinguish between stealthin C and kinamycin D. In Fig. 2a (run iv, $\Delta alp2F2G$ strain), the trace amount of kinamycin D is due to the small amount of endogenous HNO₂ which was also mentioned in our previous report (<https://doi.org/10.1021/acs.jnatprod.7b00652>).

Figure Response 3. UV absorbance spectra of kinamycin D and stealthin C.

2. In Line 185, the authors mentioned the formation of new product 4 but not clearly labeled in Fig. 2b. Even if it has been confirmed to be the same as prekinamycin, the authors should provide a label such as “4 = prekinamycin” in the figure. Identical UV absorbance spectrum has been mentioned which should be shown in SI.

Thanks for your suggestion and the new peak (compound 4) has been labeled. Also, the UV absorbance spectrum of compound 4 and prekinamycin has been added in Fig. S10a.

Fig. 2b. HPLC profiles of the crude extracts from the *E. coli* BL(21) derivatives fed with gluN_2H_3 .

Fig. S10a. UV absorbance spectrum of compound **4** and prekinamycin.

3. The authors should provide more Ctrl for the one-pot reaction shown in Fig. 3a. There are multiple enzymes (AlpJ/K/H) and cofactors (FAD, NADH) in the system. Please provide HPLC profiles for Ctrl reactions eliminating each component in SI. Thanks for your great suggestion. The HPLC profiles for control reactions eliminating each component have been provided in supplemental information (Fig. S12).

Fig. S12. HPLC analysis of the reactions with different combinations of enzymes and cofactors for the one-pot reaction.

4. Since there is time-course assay presenting in Fig. 3b, the authors should also mention the incubation time for the one-pot reactions shown in Fig. 3a. There are different amounts of compound 1 and 6 in “-gluN₂H₃” and “Boiled AlpH” traces shown in 3a and 3b. Is this caused by different incubation time, or different sample treatment methods (solvent extraction vs. direct injection)?

Thanks for your comments. The incubation time for the one-pot reactions shown in Fig. 3a was described in the revised manuscript. Different amounts of compound 1 and 6 in “-gluN₂H₃” and “Boiled AlpH” traces shown in 3a and 3b is mainly caused by different sample treatment methods. 3a: In the end of the reaction, an equal volume of ethyl acetate with 1% glacial acetic acid was used to extract the product twice. The organic extract was dried by rotary evaporation and the final extract was dissolved in 50 μ L DMSO, then 25 μ L was directly analyzed by HPLC. 3b: For the direct analysis of 5 in the reaction mixture (200 μ L), half a volume of cold methanol was added to the reaction mixture in the end of the reaction and 30 μ L of the supernatant was directly analyzed by HPLC after centrifugation. In the direct analysis (3b), more 6 was observed. During the organic extraction, 6 is not stable and converted to 1.

Fig. 3. *In vitro* functional characterization of AlpH.

5. There is no need to refer to compound 5 (fig 3b) as X. Remove this X label.
Correct as suggested.

6. There are many mistakes in Fig. 4a. The structures of prekinamycin, hydroquinone-kinobscurinone, kinobscurinone are not consistent with Fig 1. Please double check the mechanisms carefully.

We thank the reviewer for pointing out the errors in Fig. 4a. Correct as suggested.

Fig. 4a. Mechanistic alternatives for the coupling of gluN₂H₃ and polyketide intermediate.

7. Please further clarify the stepwise assays listed in Fig. 4c. The current labels are very confusing.
Correct as suggested.

Fig. 4c. HPLC analysis of the two-step reactions.

8. Line 375, the physiological function of AlpK remains to be elucidated? This is confusing as all in vitro assays in this manuscript use AlpK.

We thank the reviewer for this important comment. We deleted this sentence and added more information about the reported function of AlpK in the introduction section between line 71 and 79: During the assembly of the benzofluorene core, two FAD-dependent enzymes, AlpJ and AlpK, were identified to be involved in the contraction of the B ring in the kinamycin biosynthesis. AlpJ belongs to a unique family of FAD-dependent oxygenase and has been shown to catalyze a Bayer-Villiger-type oxidative C-C bond cleavage of DHR, forming of an unstable aldehyde/acid intermediate, which undergoes decarboxylative ring closure and dehydration to give the benzofluorene ring. AlpK could possibly supply necessary FADH₂ for AlpJ catalyzed reaction. And it was also proposed that AlpK might catalyze the C5-hydroxylation reaction of the benzo[*b*]fluorene intermediate.

9. Please further proofread the entire manuscript to correct grammar errors. In addition, more refs are needed. For example, line 408.

Correct as suggested.

Reviewer #3 (Remarks to the Author):

Diazo are promising natural compounds. In this manuscript, the authors focused on the biosynthesis of kinamycin and the mechanism for the installation of the hydrazine group onto the kinamycin scaffold. They discovered an unprecedented O-methyltransferase-like protein, AlpH, responsible for the hydrazine incorporation in kinamycin biosynthesis. Biochemical and structural characterizations of AlpH have been performed and support the critical role of AlpH in a diazo NP biosynthesis. This manuscript brings novel insights in the biosynthesis of diazo compounds and regarding SAM-independent OMT-like enzyme. I recommend publication after major revision.

Although AlpW is required for kinamycin biosynthesis, its role is unclear. Is AlpW conserved in biosynthesis gene cluster? What is the assigned function of AlpW in database? This should be discussed. In the $\Delta alpW$ strain, complementation experiment should be performed to support the critical role of AlpW.

We thank the reviewer for the important suggestion and the conservation and proposed function of Alp1W have been described in the revised manuscript. Also, the complementation experiment was performed in the $\Delta alpW$ strain (Fig. S2).

Fig. S2. *In vivo* characterization of Alp1W.

HRMS analysis should be provided for all products shown in HPLC analysis to confirm their identity.

All HRMS data for all products shown in HPLC analysis have been provided in supplemental information as suggested.

It should be discussed in which biosynthesis pathways are found AlpH homologues (Fig S16) and their protein sequence/structure/active site homology.

Correct as suggested.

The docking model is interesting and provides a rational for SAM-independent activity of AlpH. However, the structure of AlpH with the substrate (gluN₂H₃ and/or a polyketide intermediate) would further support the role of AlpH. Substrate-enzyme structure and/or binding affinity experiment should be performed.

We are grateful for the comment. We failed to get the co-crystals of AlpH and gluN₂H₃ so that we tried to study the interaction between AlpH and gluN₂H₃ by surface plasmon resonance (SPR) (Fig. S25).

Fig. S25. Surface plasmon resonance measurement of binding affinity between AlpH and gluN₂H₃. K_{on} (1/ms)= 1.01×10^5 , K_{off} (1/s)= 1.15×10^{-2} , K_d (M)= 1.14×10^{-7} .

REVIEWERS' COMMENTS

Reviewer #1 (Remarks to the Author):

The authors have properly responded to reviewers' request and improved their paper as suggested. Therefore, this paper should be accepted.

Reviewer #2 (Remarks to the Author):

The authors have provided reasonable supplementary data and addressed most of the concerns and questions from the reviewers. Authors have done an overall excellent revision and I only have a few more minor comments here:

1. The data shown in Figure Response 2 can be included in the manuscript (SI) to provide evidence that compound 7 is likely formed by AlpJK and could be a substrate of AlpH.
2. In Figure S12, please clarify the reason of including Fre in the experiments in the legend and cite the related reference Reviewer 1 mentioned so that the readers know the rationale of designing such experiments.
3. In Figure S24, the authors proposed H267 as the proton donor to protonate the keto group of proposed intermediate 9. But from their docking model, it doesn't seem that this residue is adjacent to the corresponding carbon in the B-ring. It's difficult to tell from 2D figure, but H270 seems to be more close to that carbon? Why the authors propose H267 as the proton donor? What are the distances of these residues with the corresponding carbon?
4. K266D seems to be the only single mutant that can significantly abolish the activity of AlpH, do the authors have any proposal for the reason of such effect? Is this only caused by a significant interruption of substrate binding or it affects the catalysis? Did the authors try to search for any potential intermediate or shunt products in the assays with this mutant?

Reviewer #3 (Remarks to the Author):

The authors have addressed the issues and my questions, and the current version of the manuscript has been improved. The manuscript is recommended for publication.

Complete listing of the reviewer comments and our responses

- Reviewers' comments, black.
- Our responses, blue.

Reviewer #1 (Remarks to the Author):

The authors have properly responded to reviewers' request and improved their paper as suggested. Therefore, this paper should be accepted.

We appreciate the positive comments from the reviewer.

Reviewer #2 (Remarks to the Author):

The authors have provided reasonable supplementary data and addressed most of the concerns and questions from the reviewers. Authors have done an overall excellent revision and I only have a few more minor comments here:

1. The data shown in Figure Response 2 can be included in the manuscript (SI) to provide evidence that compound 7 is likely formed by AlpJK and could be a substrate of AlpH.

Thanks for your suggestion. The figure has been included in the manuscript (SI) as suggested.

2. In Figure S12, please clarify the reason of including Fre in the experiments in the legend and cite the related reference Reviewer 1 mentioned so that the readers know the rationale of designing such experiments.

The reason of including Fre in figure S12 and the related reference have been added as suggested.

3. In Figure S24, the authors proposed H267 as the proton donor to protonate the keto group of proposed intermediate 9. But from their docking model, it doesn't seem that this residue is adjacent to the corresponding carbon in the B-ring. It's difficult to tell from 2D figure, but H270 seems to be more close to that carbon? Why the authors propose H267 as the proton donor? What are the distances of these residues with the corresponding carbon?

We thank the reviewer for this important comment. In the docking model, the distances between these two histidines and the corresponding carbon in the B-ring are 6.3 Å (H267) and 5.5 Å (H270), respectively, suggesting that both H267 and H270 could be the potential general acid to protonate the keto group of proposed intermediate 9. As the structure of AlpH including the sidechains of the residues in the active site is fixed

in the docking experiment, a little conformational change of H267 and H270 when the substrates enter the active site during the catalysis would significantly change the distances measured in the docking model. The docking result suggest that both H267 and H270 could be the potential general acid to protonate the keto group of proposed intermediate 9. In addition, the activity assays of the single mutants indicated that the activity of H267A is lower than that of H270A. Thus, we proposed H267 as the proton donor to protonate the keto group of proposed intermediate 9. However, it is hard to determine which histidine is the real proton donor based on the available data.

4. K266D seems to be the only single mutant that can significantly abolish the activity of AlpH, do the authors have any proposal for the reason of such effect? Is this only caused by a significant interruption of substrate binding or it affects the catalysis? Did the authors try to search for any potential intermediate or shunt products in the assays with this mutant?

Thanks for your comments. It is interesting that K266D is the only single mutant that can significantly abolish the activity of AlpH among the 7 single mutants in this study. We reason that the Mannich reaction has high thermodynamic tendency and the key catalytic role of AlpH is to bring the two substrates (7 and gluN₂H₃) together and in suitable conformations. In our docking model, it was suggested that K266 is important for the interaction of AlpH with the carboxyl group of gluN₂H₃. Thus, K266D mutation severely interrupt the gluN₂H₃ binding and destroy the activity of AlpH. Only 1 and 6 (shunt products) could be detected in the assays with the K266D mutant.

Reviewer #3 (Remarks to the Author):

The authors have addressed the issues and my questions, and the current version of the manuscript has been improved. The manuscript is recommended for publication.

We appreciate the positive comments from the reviewer.